# Dealing with non-stationarity in sub-daily stochastic rainfall models

Lionel Benoit[1], Mathieu Vrac[2], and Gregoire Mariethoz[1]

[1]Institute of Earth Surface Dynamics (IDYST), University of Lausanne, Lausanne, Switzerland
[2]Laboratory for Sciences of Climate and Environment (LSCE-IPSL), CNRS/CEA/UVSQ, Orme des Merisiers, France

**Correspondence:** Lionel Benoit (lionel.benoit@unil.ch)

**Abstract.**

Understanding the stationarity properties of rainfall is critical when using stochastic weather generators. Rainfall stationarity means that the statistics being accounted for remain constant over a given period, which is required for both inferring model parameters and simulating synthetic rainfall. Despite its critical importance, the stationarity of precipitation statistics is often regarded as a subjective choice whose examination is left to the judgement of the modeler. It is therefore desirable to establish quantitative and objective criteria for defining stationary rain periods. To this end, we propose a methodology that automatically identifies rain types with homogeneous statistics. It is based on an unsupervised classification of the space - time - intensity structure of weather radar images. The transitions between rain types are interpreted as non-stationarities.

Our method is particularly suited to deal with non-stationarity in the context of sub-daily stochastic rainfall models. Results of a synthetic case study show that the proposed approach is able to reliably identify synthetically generated rain types. The application of rain typing to real data indicates that non-stationarity can be significant within meteorological seasons, and even within a single storm. This highlights the need for a careful examination of the temporal stationarity of precipitation statistics when modelling rainfall at high resolution.

## 1 Introduction

Stochastic rainfall models are statistical models that aim at simulating realistic random rains. For this purpose, they generate rainfall simulations which reproduce, in a distributional sense, a set of key rainfall statistics derived from an observation dataset (Benoit and Mariethoz, 2017). The practical interest of stochastic rainfall models is notably to complement numerical weather models for the simulation of rainfall heterogeneity at fine scales, and to quantify the uncertainty associated with rainfall reconstructions. Indeed, numerical weather models face challenges for reproducing rainfall heterogeneity in space and time, in particular at fine scales (Bauer et al., 2015; Bony et al., 2015). Some of the main applications of stochastic rainfall models are therefore the fast generation of synthetic rainfall inputs for local impact studies related for instance to hydrology (Paschalis et al., 2014; Caseri et al., 2016) or agronomy (Mavromatis and Hansen, 2001; Qian et al., 2011), and the downscaling of aggregated precipitation products such as rain observations (Allcroft and Glasbey, 2003; Bárdossy and Pegram, 2016) or numerical model outputs (Wilks, 2010; Vaittinada Ayar et al., 2016). In all cases, the target is the transposition of observed rain statistics into synthetic rain simulations.

Recently, a considerable attention has been paid to increasing the resolution of stochastic rainfall models so that they can mimic

rainfall at sub-daily time scales. Currently, several high resolution stochastic rainfall models are able to deal with precipitation data at typical resolutions of 1 min to 1 h in time and of 100 x 100 m$^2$ to 1 x 1 km$^2$ in space (see e.g. (Leblois and Creutin, 2013; Paschalis et al., 2013; Benoit et al., 2018). At such scales, not only the marginal distribution of observed rain intensity matters, but the space-time dependencies within rain fields are also important features of the rain process (Emmanuel et al.,

2012; Marra and Morin, 2018). In particular, the impact of the advection and diffusion of spatial rainfall patterns (e.g. rain cells or rain bands) have to be modelled (Lepioufle et al., 2012; Creutin et al., 2015). In consequence, most sub-daily stochastic rainfall models consider rainfall as a space-time random process.

An underlying hypothesis in stochastic rainfall modelling is that of stationarity: the statistics of rainfall are supposed to be constant over a given (space-time) modelling domain. This enables (1) the inference of rainfall statistics from an observation

dataset, and (2) the reproduction of these statistics in simulations. The definition of stationary domains can be regarded as a modelling choice, often subjective and left to the judgment of modelers (Journel, 1993). It consists of defining pools of data that are considered similar enough (in a statistical sense) to perform model inference. In the case of stochastic rainfall modelling, the identification of stationary datasets or sub-datasets relies on some phenomenological guesses about rainfall, which serve as fuzzy guidelines to delineate stationary domains. Depending on the application and modelling choices, the parametrization of

sub-daily stochastic rainfall models is considered as changing at scales ranging from seasons (Paschalis et al., 2013; Bárdossy and Pegram, 2016; Peleg et al., 2017) to single rain storms (Caseri et al., 2016; Benoit et al., 2018).

One possible approach to delineate pools of homogeneous rain observations in a more quantitative way is to classify them prior to modelling. A set of predefined criteria is used to build a metric of similarity between the observations, and a classification algorithm is applied to the resulting similarity measures in order to define clusters of closely related rain observations. The

result of such a classification procedure, often referred to as rain typing, is the identification of a limited number of rain types which gather rain observation sharing similar properties. Until recently, rain typing mainly focused on classifications based on rain intensity only, with the aim to assess the physical processes responsible of rain generation (e.g. distinguish convective and stratiform rains) (Rosenfeld and Amitai, 1995; Biggerstaff and Listemaa, 2000; Llasat, 2001). In the last years, the emergence of metrics characterizing rainfall spatial or space-time behavior (Vrac et al., 2007; Ramirez-Cobo et al., 2010; Aghakouchak

et al., 2011; Zick and Matyas, 2016) paved the way to new rain typing methods based on the arrangement of rain fields in space and in time (Leblois, 2012; Lagrange et al., 2018).

In this context, the present paper focuses on the temporal non-stationarity of rainfall space-time statistics in view of sub-daily stochastic rainfall modelling. We restrict intentionally our investigation to temporal non-stationarities, and consider stationarity in space (i.e. constant statistics over the whole area of interest) as a prerequisite modelling assumption. The goal is therefore

to identify periods of time during which rainfall space-time statistics remain as constant as possible over a given area. The proposed framework relies on the classification of radar images based on their space-time features. The resulting classes are then used to define rain types that group rain fields with similar statistical signatures. Finally, the transition between rain types is interpreted as a break in the temporal stationarity of rainfall statistics.

The remainder of this paper is structured as follows: Sect. 2 gives a general overview of rainfall space-time patterns visible in

radars images. Sect. 3 describes a rain typing method based on the previously identified patterns, and explains how the resulting

rain types can be used to identify stationary periods. Then, Sect. 4 assesses the performance of this method for both synthetic and real case studies. Next, Sect. 5 discusses the dependence of the proposed rain typing method to the stochastic model in use, as well as the implications of the observed patterns of non-stationarity for sub-daily stochastic rainfall modelling. Finally, Sect. 6 gives general conclusions.

## 2 Overview of rainfall space-time patterns observed in radar images

Prior to the design of a quantitative method to identify non-stationarities in rainfall statistics, the current section seeks to illustrate with some typical examples the diversity of space-time patterns that can be observed in rain fields, and to give an overview of their temporal evolutions.

We illustrate this study with data collected over the Vaud Alps, Switzerland (Fig 1a). The area of interest encompasses a network of high-resolution rain gauges used later for validation, and covers an area of 60 x 60 km$^2$. For reasons of data availability, we focus in this paper mostly on summer rains observed from 2017 July, 1 to 2017 August, 31. During this period, only periods corresponding to rain events are considered. A rain event is defined as a rainy period isolated by at least 30 min of dry conditions before and after, and resulting in at least 2 mm of cumulated rain height (in average over the area of interest). The dataset is comprised of 17 rain events causing around 250 mm of cumulative rain height.

Fig. 1b-c displays different rain fields observed by the Swiss weather radar network operated by the MeteoSwiss weather agency (Germann et al., 2006). Weather radars are remote sensing devices providing comprehensive images of rain fields at the regional scale (coverage up to about 200 km from the radar device), with a high resolution (in the present case, 10 min in time and 1 km in space). The resulting rain rate estimates are known to be biased (Berndt et al., 2014), but in counterpart radar images are currently the most reliable and exhaustive source of information about the spatial organization of rain fields and their temporal evolution (Emmanuel et al., 2012; Thorndahl et al., 2017; Marra and Morin, 2018): this is why radar images are used in the current section to illustrate rainfall space-time behavior, and in the following to extract rainfall statistics.

A visual inspection of the rain fields displayed in Fig. 1 allows understanding the interest of characterizing jointly the spatial and the temporal behavior of high resolution rain fields. At the scale of interest, all rain fields shown in Fig. 1b-c have in common strong space-time interactions structured by:

- A distribution of intensities that is often skewed (Vrac and Naveau, 2007), with a variable amount of zero values due to within-storm rain intermittency (Schleiss et al., 2011; Mascaro et al., 2013).

- Well defined spatial patterns (Guillot, 1999; Emmanuel et al., 2012; Marra and Morin, 2018), which can be linked to the processes responsible for rainfall production such as rain cells, rain bands or rain storms.

- A temporal behavior shaped by the advection and diffusion of spatial patterns along time (Lepioufle et al., 2012; Creutin et al., 2015).

Despite common space-time attributes, the three rain events of Fig. 1b look very different. For example, the Type A event is characterized by a strong spatial intermittency (i.e. dry and wet locations coexist in the same radar image) combined with

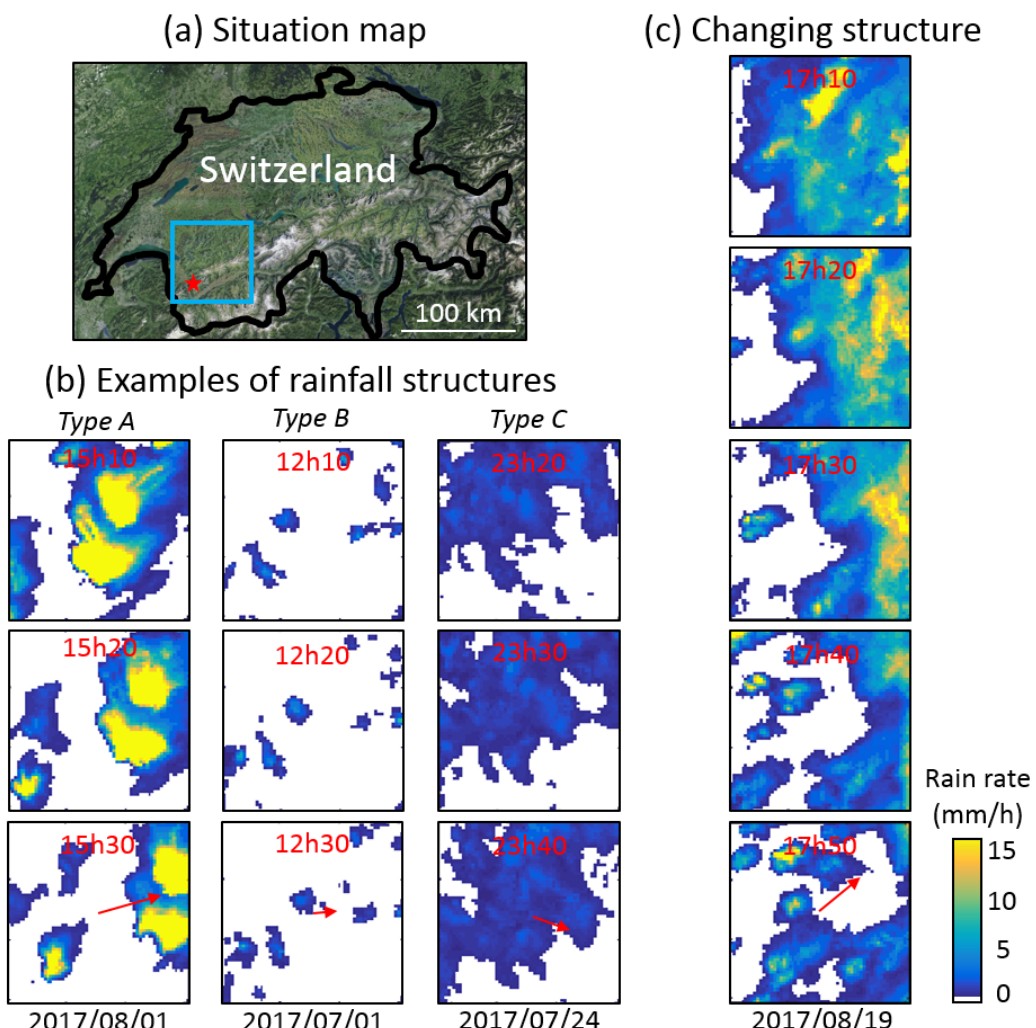

**Figure 1.** Examples of rain fields over Vaud Alps, Switzerland. (a) Situation map. The area of interest is delineated by the blue square. The red star denotes the location of the rain gauge network used for validation (Sect. 4). (b) Examples of rain storms with different space-time behaviors. (c) One example of rain field with a temporally changing behavior. In (b) and (c) the red arrows displayed in the last panels denote the advection of the rain storm in 20 min.

well-defined areas generating intense rainfall, while the Type C event shows a lower fraction of dry locations, is less spatially structured and generates low rain rates. Hence, it should be possible to find statistical metrics able to distinguish these three rain types based on their spatio-temporal characteristics.

It is worth noting that not only the space-time statistics of rainfall change between rain events, as shown in Fig. 1b, but also these statistics can change within a single rain event, as illustrated in Fig. 1c. In this case, a widespread and spatially continuous rain field (2 first images) is replaced by disconnected rain aggregates (2 last images). This change in the space-time features is

very rapid and takes place in less than 30 min. Such abrupt changes in the space-time behavior within a rain event are relatively common in our dataset as discussed later.

Starting from this example, this paper investigates how to detect non-stationarities in rainfall space-time statistics using radar images as primary information. We adopt an Eulerian approach and investigate the temporal variability of rainfall statistics over a given area of interest, as perceived by an Earth-fixed observer.

## 3   Assessing rain statistics stationarity from radar images

### 3.1   Extracting space-time information from radar images

To assess the stationarity of rainfall space-time statistics, we propose to start by extracting information on the rainfall space-time behavior from radar images. To this end, 10 statistical metrics are derived for every radar image (Fig. 2), which are split in three categories that reflect the three main characteristics of rain fields identified in Sect. 2:

- Intensity Indices (II), which relate to the probability density function (histogram) of the rain intensities measured in a given radar image. The following indices are used:

  - II.1: Fraction of the image covered by rainy pixels (informs the intra-storm rain intermittency).
  - II.2: Mean rain intensity computed over all rainy pixels.
  - II.3: Quantile 80% of rain intensities characterizing heavy rain pixels.

- Spatial Indices (SI) that characterize the spatial arrangement of patterns within rain fields. They are selected among the indices proposed by (Aghakouchak et al., 2011). They are computed based on binary images representing rain masks (Fig 2). Such binary images are obtained by thresholding radar images at a rain intensity of 0.1mm/h, and by assigning a 0 value to the pixels under the threshold and a 1 value otherwise. Then, connected components, hereafter referred to as rain aggregates, are identified in every binary image. Their morphological properties are used to derive the following indices:

  - SI.1: Fraction of rainy area covered by the largest rain aggregate in the image. This is a first indication of how the rain field is split into aggregates. Let $N_p$ be the total number of rainy pixels in the binary image and $N_m$ be the number of pixels of the largest aggregate; then $SI.1 = \frac{N_m}{N_p}$.
  - SI.2: Connectivity index. It is equal to one if the rain field is fully connected (one single rain aggregate) and tends to zero if the rain field is split into many disconnected aggregates. Let $N_c$ be the number of rain aggregates in the binary image, then: $SI.2 = 1 - \frac{N_c - 1}{\sqrt{N_p + N_c}}$.
  - SI.3: Perimeter index, characterizing the sinuosity of the contours of rainy areas. It is equal to 1 if all rain aggregates are squares, and tends to 0 if the rain aggregates are very sinuous. Let $p$ be the total perimeter of rain aggregates, i.e. the sum of the perimeters of all rain aggregates, then:
  
  $$SI.3 = \frac{4 \times \sqrt{N_p}}{p} \; if \; \lfloor \sqrt{N_p} \rfloor = \sqrt{N_p}$$
  $$SI.3 = \frac{2 \times (\lfloor 2 \times \sqrt{N_p} \rfloor + 1)}{p} \; if \; \lfloor \sqrt{N_p} \rfloor \neq \sqrt{N_p}$$
  .

- SI.4: Area index, characterizing the spread of the rain aggregates. It is equal to 1 if the radar image contains one single aggregate, and tends to zero if the rainy pixels are only in the corners of the image. Let $A_{convex}$ be the area of the convex hull encompassing all the rain aggregates, then: $SI.4 = \frac{N_p}{A_{convex}}$.

- Temporal Indices (TI), which characterize the temporal evolution of the rain fields. They assess the advection of rain storms over the ground as well as the temporal deformation of spatial rain patterns. Let $I_t$ and $I_{t+1}$ be two subsequent images. In addition, let $r_{i,j}$ be the cross-correlation between $I_{t+1}$ and $I_t$ translated by a vector $\overrightarrow{D} = i.\overrightarrow{E} + j.\overrightarrow{N}$ of coordinates $i$ and $j$ along the Eastward and Northward directions respectively. Finally, let $r^{max}$ be the maximum correlation and $\overrightarrow{D^{max}} = i^{max}.\overrightarrow{E} + j^{max}.\overrightarrow{N}$ the corresponding displacement vector. Then the Temporal Indices (TI) are defined by:

  - TI.1: Eastward component of the displacement vector, i.e. $i^{max}$. This index corresponds to the advection of the rain storm along the West-East direction between times $t$ and $t+1$.

  - TI.2: Northward component of the displacement vector, i.e. $j^{max}$. This index corresponds to the advection of the rain storm along the South-North direction between times $t$ and $t+1$.

  - TI.3: Correlation coefficient between the two radar images $I_t$ and $I_{t+1}$ after removing advection effects, i.e. $r^{max}$. This index equals one if the spatial rain patterns remain identical between two subsequent radar images (up to a translation), and tends to zero if the images are completely different.

### 3.2 Classification of radar images based on rainfall space-time statistics

The 10 indices defined above are used to classify the radar images in order to obtain a limited number of rain types. To ensure the reliability of these indices, only images with a significant proportion of rainy pixels are used for classification. Indeed, if the number of rainy pixels is low, the space indices (SI) are not meaningful and the time indices (TI) cannot be computed because the image correlation procedure fails. We therefore only classify the images with more than 10% rainy pixels, the remaining rainy images (rain fraction <10%) being typed afterwards.

To define rain types, we adopt an approach based on a Gaussian Mixture Model classifier (GMM) (Fraley and Raftery, 2002; Pernin et al., 2016). This classifier has been selected because it allows for an automatic selection of the number of classes, and because it does not require any a priori information about the joint distribution of the rain indices. The idea of the GMM is to approximate the joint distribution of the 10 statistical indices. This approximation being a combination of Gaussian functions, those can be considered as representing several discrete categories. In the GMM, the joint distribution $\hat{p}(\boldsymbol{x})$ of the space-time indices forming a vector $\boldsymbol{x} \in \mathbb{R}^{10}$ is approximated by a weighted sum of $K$ multivariate Gaussian distributions $\mathcal{N}(\boldsymbol{x}|\mu_k, \Sigma_k), k = 1, ..., K$, with respective mean vector $\mu_k$ and covariance matrix $\Sigma_k$:

$$\hat{p}(\boldsymbol{x}) = \sum_{k=1}^{K} \pi_k \times \mathcal{N}(\boldsymbol{x}|\mu_k, \Sigma_k) \tag{1}$$

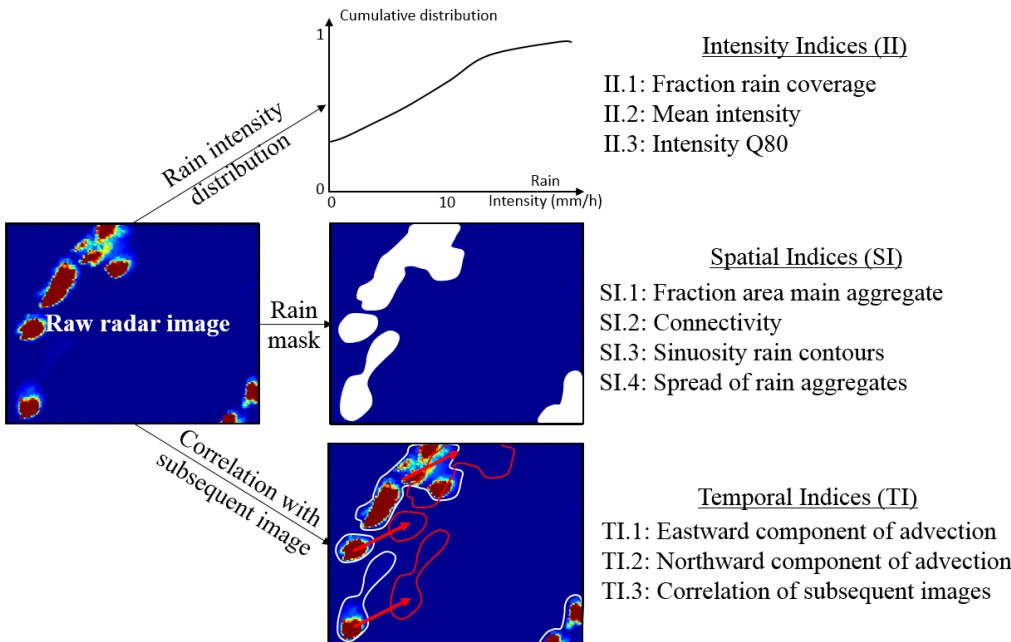

**Figure 2.** Computation of indices characterizing rainfall space-time statistics for a single radar image. This procedure is repeated for each image with >10% rainy pixels. Note that the temporally subsequent image is required to compute the Time Indices (TI).

The inference of the model parameters (i.e $\pi_k, \mu_k, \Sigma_k, k = 1, ..., K$) is performed with an Expectation-Maximization (EM) algorithm (Fraley and Raftery, 2002). A full covariance model is used for the covariance matrices $\Sigma_k$ in order to take into account a possible correlation between the indices. The number $K$ of Gaussian mixtures used in the GMM model is selected by minimization of the BIC criterion derived from EM fits computed for different numbers $K$ (Schwartz, 1978), with the goal of selecting the GMM model resulting in the best fit while maintaining a parsimonious parametrization. Here the Matlab Statistics and Machine Learning Toolbox has been used to fit the GMM model, with the function 'fitgmdist'.

Once fitted, the GMM can be used to derive a probabilistic classification of any vector $\boldsymbol{x}$ of indices. The probability that a vector belongs to the population $G_j$ whose distribution is the $j^{th}$ mixture component $\mathcal{N}(\boldsymbol{x}|\mu_j, \Sigma_j)$ is given by:

$$\hat{p}(\boldsymbol{x} \in G_j) = \frac{\pi_j \times \mathcal{N}(\boldsymbol{x}|\mu_j, \Sigma_j)}{\sum_{k=1}^{K} \pi_k \times \mathcal{N}(\boldsymbol{x}|\mu_k, \Sigma_k)} \qquad (2)$$

A classification of the entire image dataset can thus be obtained by assigning to each image $I$ the class that corresponds to the most probable mixture component (in practice, the Matlab 'cluster' function is used):

$$G(I) = max_j(\hat{p}(\boldsymbol{x} \in G_j)) \qquad (3)$$

As mentioned above, the classification procedure can only be applied to radar images with a significant proportion of rainy pixels (>10%). In addition, it can be desired to avoid high frequency successions of rain types, for instance if the targeted

stochastic rainfall model requires long lasting pools of data to perform parameter inference. To do this, we impose a temporal persistence threshold for the rain types. To this end, all images that lead to temporal clusters of classes that do not reach a certain duration are set to unclassified. Here we use a 60-minute duration threshold. After cleaning the classification, all the time steps that are unclassified (either because less than 10% of the pixels are rainy or because the image belongs to a <60 min

cluster) receive the type of the classified image that is temporally the closest (i.e. nearest neighbor interpolation along the time axis). The complete rain typing framework is summarized in Fig. 3.

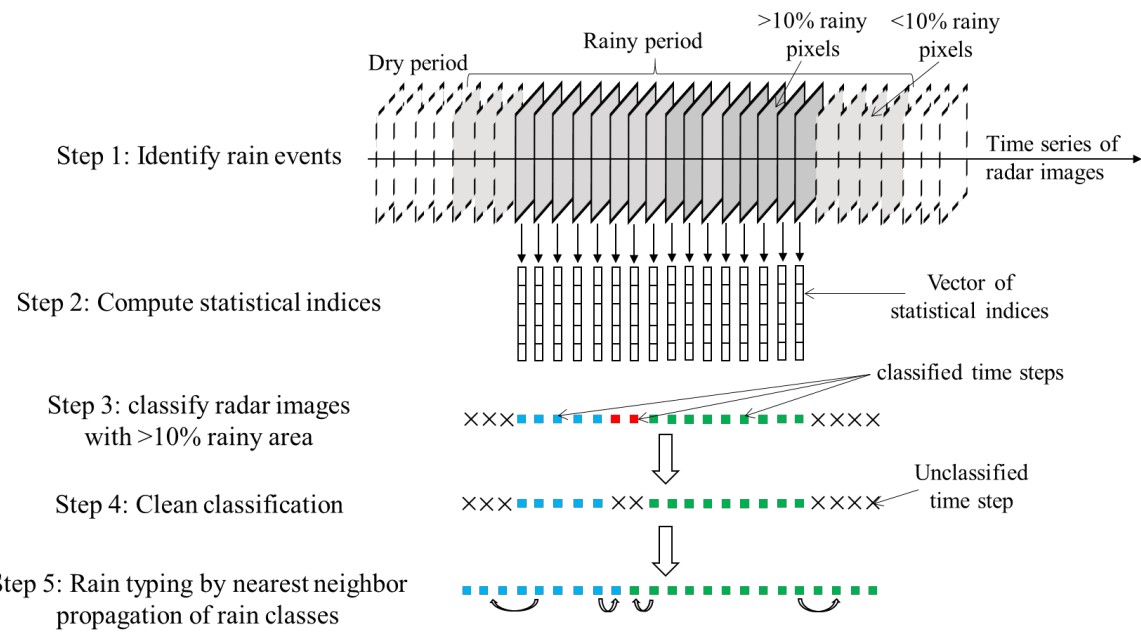

**Figure 3.** Rain typing framework.

Since the rain typing method presented above aims at defining groups of rain fields sharing similar statistical signatures, the transitions between rain types can be interpreted as non-stationarities. Similarly, periods with a constant rain type are interpreted as stationary periods.

**4   Validation and application**

In this section, we validate our rain typing approach in the context of stochastic rainfall modelling. The validation study comprises four steps: First, the proposed approach is tested in a synthetic case in order to determine whether we are able to identify known non-stationarities. Then, real data are used to compare our rain typing strategy with two alternative hypotheses of rainfall stationarity: (1) rainfall statistics are stationary at a seasonal scale and (2) rainfall statistics are stationary at a rain

storm scale. Next, the rain typing method is applied to radar data covering the full year 2017, in order to assess its ability to handle various rainfall situations. Finally, the sensitivity of the classification to the calibration dataset is assessed by comparing

rain types computed for the summer of 2017 based on two calibration periods: (1) the summer of 2017, and (2) the full year 2017. Prior to the validation itself, the next subsection describes the stochastic rainfall model used for validation.

## 4.1 Stochastic rainfall model

The validation of the rain typing approach uses a stochastic rainfall model designed for local-scale (a few square kilometer extent) and high-resolution (up to 1 min) data. This model involves 11 parameters and aims at modelling both the marginal distribution of observed rain intensities and the space-time dependencies that exist within rain fields. It is briefly introduced hereafter; for more details the reader is referred to Benoit et al. (2018). In this model, the marginal distribution of rain rates is accounted for by considering that rain measurements ($R$) originate from the censoring and power transform (involving parameters $a_0, a_1, a_2$) of a standardized multivariate Gaussian random field ($Z$) tainted by an additive measurement noise ($\epsilon \sim \mathcal{N}(0, \sigma_\epsilon)$) (Eq. 4):

$$
R = \left( \frac{Z + \epsilon - a_0}{a_1} \right)^{\frac{1}{a_2}} \text{ if } Z + \epsilon > a_0
$$

$$
R = 0 \qquad \qquad \text{ if } Z + \epsilon \le a_0
$$

(4)

The multivariate Gaussian latent random field ($Z$) is characterized by an asymmetric Gneiting space-time covariance (Gneiting, 2002) function $\rho$ which accounts for both the advection and the diffusion of spatial rain patterns (Eq. 5). For two rain observations separated by a spatial lag $\boldsymbol{h}$ and a temporal lag $u$, the covariance is given by:

$$
\rho(\mathbf{h}, u) = \frac{1}{(u/d)^{2\delta} + 1} \exp \left( \frac{-(||\mathbf{h} + \mathbf{V}.u||)/c)^{2\gamma}}{((u/d)^{2\delta} + 1)^{\beta\gamma}} \right)
$$

(5)

In this model, the advection of rain storms is assumed to be constant and linear along a vector $\boldsymbol{V}$ defined by its amplitude $V_S$ and direction $V_\theta$. The regularity parameters $\gamma$ (for space) and $\delta$ (for time) control the slopes at the origin of the covariance function and thereby regulate the small-scale variability of the rain fields, and ultimately their smoothness. The scale parameters $c$ (for space) and $d$ (for time), in units of distance and time respectively, control the decorrelation distances of rain patterns. Finally, the separability parameter $\beta$ controls the space-time interactions. When $\beta = 0$, the covariance function is space-time separable.

## 4.2 Detection of rainfall non-stationarity in a controlled setting

The ability of the rain typing method to detect possible non-stationarities is tested by applying it to synthetic time series of radar images. These images are generated using the stochastic rainfall model presented above, with model parameters changing abruptly. This produces (temporal) non-stationary synthetic rain fields. The rain typing method is then applied to the simulated radar-like images (resolution: 1 x 1 km$^2$ in space, 10 min in time; footprint: 60 x 60 km$^2$) in order to assess if it is able to retrieve the prescribed patterns of temporal non-stationarity.

For generating the synthetic images, we use the stochastic rainfall model described in Sect. 4.1 with model parameters corresponding to three typical rain behaviors identified by visual inspection (Table 1). Fig. 4a shows an example of simulated rain field for each rain type.

|        | $a_0$ | $a_1$ | $a_2$ | $\sigma_\epsilon$ | $\gamma$ | $c$   | $\delta$ | $d$  | $\beta$ | $V_S$ | $V_\theta$ |
|--------|-------|-------|-------|-------------------|----------|-------|----------|------|---------|-------|------------|
| Type 1 | -0.73 | 0.78  | 0.41  | 0.0               | 0.76     | 4642  | 0.89     | 590  | 0.97    | 5.6   | 18         |
| Type 2 | 0.0   | 0.64  | 0.41  | 0.0               | 0.49     | 6840  | 0.86     | 892  | 0.91    | 1.2   | -11        |
| Type 3 | -0.83 | 1.16  | 0.45  | 0.0               | 0.38     | 13995 | 0.81     | 1702 | 0.95    | 0.9   | -39        |

**Table 1.** Parameters of the stochastic rainfall model used for the generation of synthetic images.

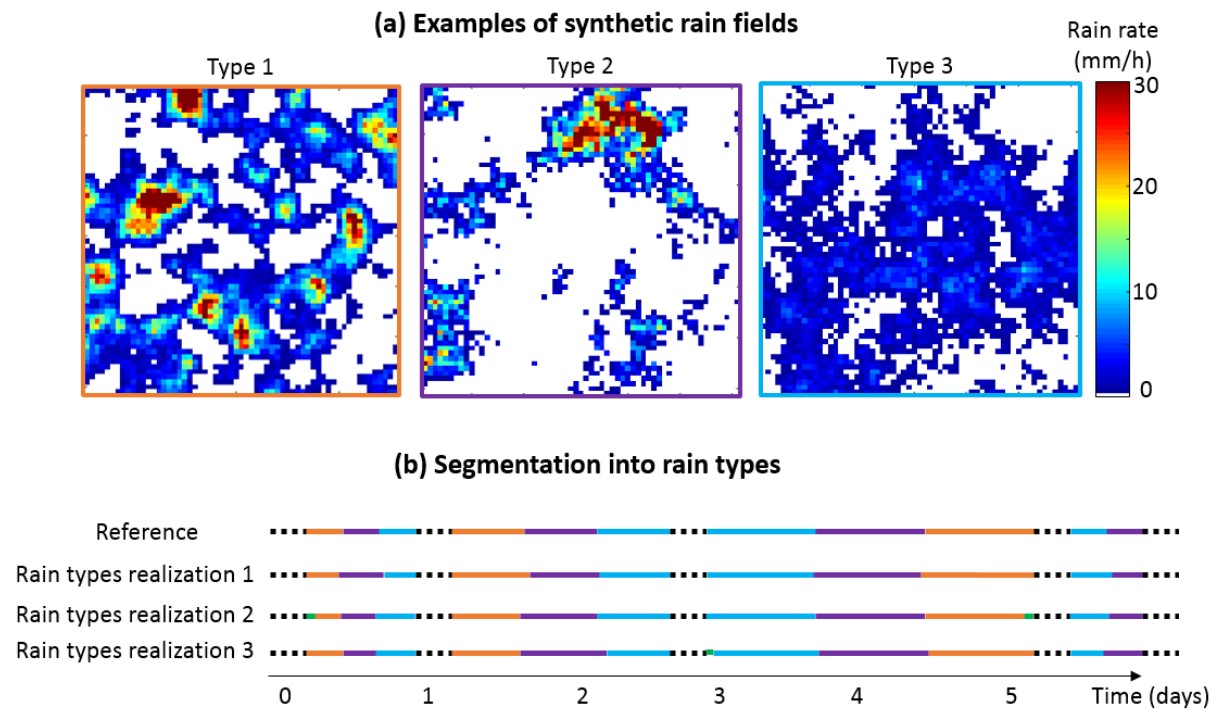

**Figure 4.** Identification of non-stationarities in rain statistics for a synthetic case study. (a) Examples of synthetic rain fields simulated for each rain type. (b) Segmentation of the time axis into periods with stationary rain statistics. First row: reference. Rows 2-4: Segmentation obtained by rain typing applied to synthetic radar-like images. Dotted black lines represent dry periods.

In Fig. 4b, the series titled "reference" shows the rain types prescribed to the stochastic rainfall model for the generation of the synthetic radar images. Based on stochastic simulations, three sets (realizations) of synthetic images are generated. Each realization is classified to determine whether it is possible to identify the reference rain types. Results show that the proposed

method can consistently detect the prescribed rain types and their temporal evolution, with an agreement between the reference and the estimated rain types of 97 %, 96.3 % and 92.6 % for realizations 1 to 3. It also properly estimates the number of rain types prescribed in the reference. The only noticeable difference between the reference and the simulations is the emergence of a very infrequent fourth rain type (accounting for 0%, 1% and 1.6% of the estimated rain types for realizations 1 to 3) at the beginning or at the end of some rain events (in green in Fig. 4b). This is because at these periods, the rain does not cover the whole area of interest, and in certain situations it can produce rain fields with different space-time statistics, which induces this artificial fourth rain type. Except for this fourth rain type, results show that in this synthetic experiment, the proposed method performs well to detect non-stationarities of rainfall space-time statistics and, in turn, periods during which these statistics remain stationary.

## 4.3    Impact of rainfall non-stationarity on stochastic modelling of an actual dataset

To further validate our rain typing method, we apply it to a real dataset acquired in the Vaud Alps, Switzerland, during the summer of 2017. In such a real case study, the true succession of rain types is obviously unknown. To assess the performance of the proposed rain typing method, we compare it with two other hypotheses of stationarity that can be found in the literature. We therefore consider three cases, illustrated in Fig. 5:

– Hypothesis H1: the time axis is broken down into rain types interpreted as stationary time periods (the approach proposed in this paper). Applying the rain typing method presented in Sect. 3 to the period of interest leads to 6 rain types.

– Hypothesis H2: the statistical signature of rainfall is constant over meteorological seasons (Paschalis et al., 2013; Bárdossy and Pegram, 2016; Peleg et al., 2017). This lead to one stationary pool of rain events for the period of interest.

– Hypothesis H3: rainfall statistics are constant within a single rain storm but changes between storms (Caseri et al., 2016; Benoit et al., 2018). Here 17 rain events are identified following the definition adopted in Sect. 2.

To compare these three hypotheses, we apply the same stochastic model as above to rain data collected by a dense network of 8 high-resolution rain gauges set up in a small (3 x 6 km$^2$, Fig. 5a) alpine catchment called 'Vallon de Nant' situated within the area of interest presented in Sect. 2. Hence, in the following, radar images will be used only to carry out the rain typing presented in Sect. 3 in order to define the hypothesis H1. The remaining of the validation, i.e. stochastic model calibration and simulation under the three tested hypotheses, will be carried out on rain rate time series acquired by rain gauges, and not on radar images. The goal is indeed to keep the following validation as independent as possible from the radar images used for rain typing. By doing so, we seek to prove that the proposed method captures the stationarity of the rainfall process itself, and not only the stationarity of radar images.

Once the periods of stationarity have been built for each of the three hypotheses, the stochastic model is calibrated for each stationary period. This means that for each hypothesis, a set of model parameters is inferred from observations for each postulated stationary dataset. Then, synthetic rain fields are generated by unconditional simulation under the three hypotheses of stationarity, and in each case 50 realizations (i.e. 50 simulated synthetic rain histories) are compared to actual measurements.

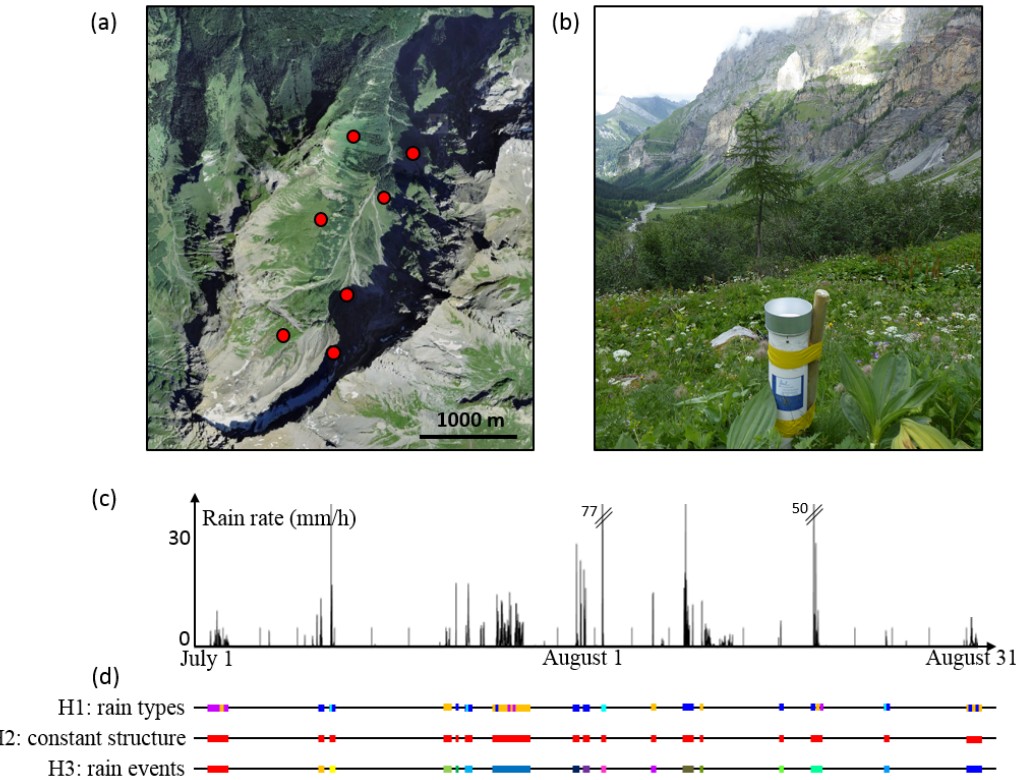

**Figure 5.** Observation dataset used for validation. (a) Measurement network. Red dots denote rain gauge locations. (b) Picture of a rain gauge with the Vallon de Nant catchment in the background. (c) Observed rain rate averaged over the network. (d) Segmentation of the time axis into stationary periods for the three tested hypotheses. For each hypothesis, segments with the same color denote periods for which rainfall is expected to have similar space-time statistics.

To assess the realism of the different scenarios, Fig. 6 shows the simulated cumulative rain heights. Next, Fig. 7 shows quantile-quantile (q-q) plots for four statistics selected to assess the marginal distribution of rain rates and its space-time arrangement: number of rain gauges measuring zero rain at each time step, rain intensity, standard deviation (in time) of rain intensities separated by a time lag of 5 min, and standard deviation (in space) of rain intensities at each time step.

5    Results show that H1 tends to slightly underestimate the cumulative rain due to an underestimation of very high intensities. This underestimation of heavy rainfalls is common to all the three cases, and probably originates from the stochastic model itself, which is not designed to handle extreme rainfalls due to the simple transform function selected in Eq. 4. This could be improved by adopting a transform function accounting for extreme rainfalls (see e.g. (Vrac and Naveau, 2007)) but at the price of a more complex parametrization which is not regarded as essential here because the observed rain rates are mostly low to

10    moderate, and only the 99th centile is affected by rain rate underestimation. Apart for this underestimation of high rain rates, hypothesis H1 allows correctly reproducing the other metrics.

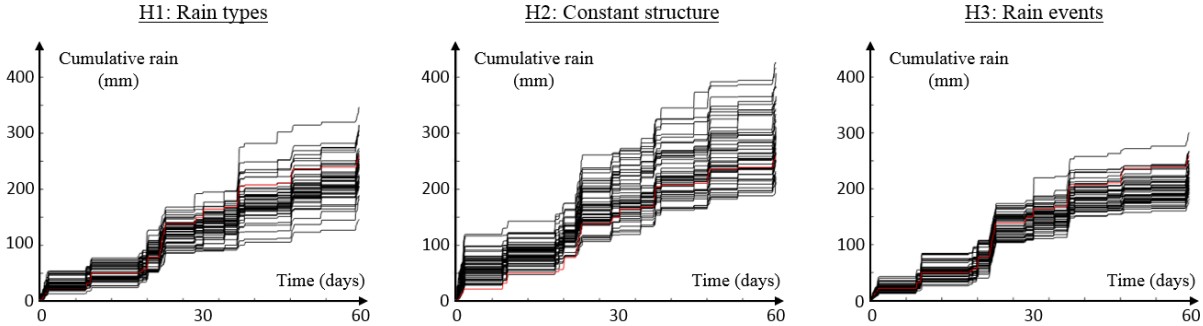

**Figure 6.** Reproduction of the cumulative rain height (averaged over the whole network) for the three tested hypotheses. Left: rain is stationary by rain type (H1), Center: rain is stationary during the entire summer period (H2), Right: rain is stationary by rain event (H3). The red line corresponds to the observations and the black lines correspond to simulations.

On the contrary to H1, hypothesis H2 leads to a slight overestimation of the simulated rain height, in particular for the first 30 days (Fig. 6). This is due to the overestimation of moderate rain rates that compensates for the underestimation of extremely high values. This bias in the simulated marginal distribution is due to the lack of flexibility of H2 that imposes a single underlying stochastic model for the entire summer period. This does not allow enough flexibility to capture the diversity of structures

emerging from high-resolution data. This is also visible for the simulated variability in space and time, which tends to be overestimated for the low centiles but underestimated for the high centiles.

Under hypothesis H3, simulation results are close to the ones of H1, with a greater propensity of underestimating heavy rainfalls. In addition, the standard deviation in time is not perfectly reproduced for the middle quantiles. This slightly lower performance of H3 compared to H1 can be attributed either to the non-inclusion of intra-storm non-stationarities in this hy-

pothesis leading to a poorer reproduction of the true rainfall dynamics, or to a poor inference of model parameters in case of short rain events caused by the low amount of observations available for such very short stationary periods.

To sum up, the proposed method consisting in typing rain fields according to their space-time statistical signature derived from radar images (H1) leads to more realistic rainfall simulations than the other approaches H2 and H3.

### 4.4   Seasonality of rain type occurrence

The two previous sections have shown that the proposed rain typing method is able to reliably identify rain types when applied to summer rains. To complement the previous findings, the current section investigates the ability of our rain typing framework to classify rainfall from other seasons. To this end, rain typing is performed for the same area of interest as above (Fig 1a), but this time for a complete year of radar observations. Here no comparison data are available since the rain gauge network used for validation in Sect. 4.3 had to be removed after the summer due to local harsh winter conditions. For this reason, this section

concentrates on a qualitative analysis of the timing of rain type occurrences throughout the year, with interpretation of some resultant rain types.

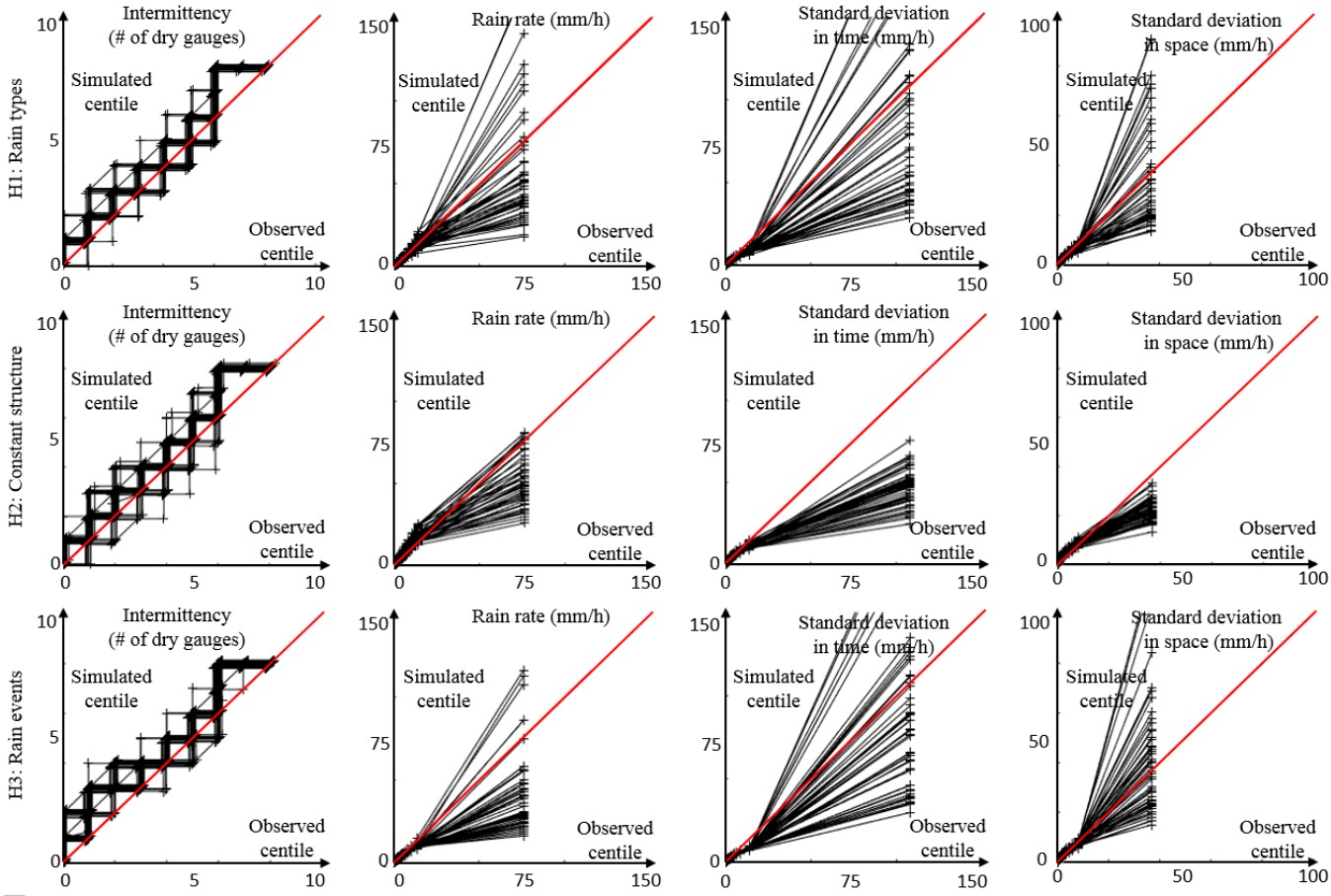

**Figure 7.** Reproduction of rainfall statistics for the three tested hypotheses. Row 1: rain is stationary by rain type (H1), Row 2: rain is stationary during the entire summer period (H2), Row 3: rain is stationary by rain event (H3). Column 1: quantile-quantile (q-q) plot of simulated vs observed rain intermittency, Column 2: q-q plot of simulated vs observed rain rate, Column 3 q-q plot of simulated vs observed temporal variability of rainfall (at lag 5 min), Column 4: q-q plot of simulated vs observed spatial variability of rainfall. The quantiles used in the q-q plots are centiles. Each centile is denoted by a black cross.

The classification framework presented in Sect. 4.3 is applied to radar data covering the entire year 2017. It should be noted that rainfall and snowfall are processed without distinction, and thus the resulting classification produces precipitation types rather than rain types strictly speaking. However, to be coherent with the rest of the paper, we will continue to refer to rain types despite possible mixes between rain and snow during the winter months.

5   The automatic selection of the number of Gaussian mixtures used in the GMM model leads to 11 rain types for this year. Fig. 8 shows the monthly occurrence of these rain types, and Fig. 9 shows the related marginal distributions of the 10 indices used to characterize rainfall space-time behavior.

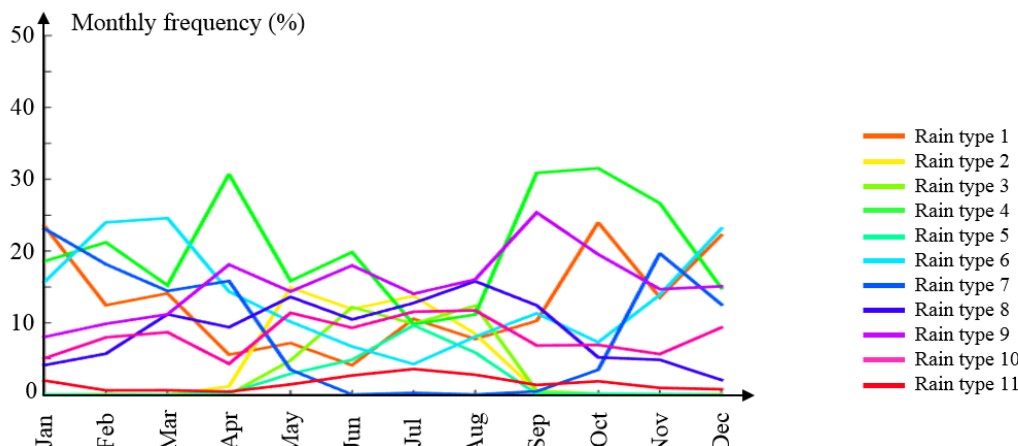

**Figure 8.** Monthly occurrence of rain types during the year 2017.

The monthly occurrence of rain types shows a clear seasonality for most of rain types (Fig. 8) with some types occurring mostly during winter months (types 1, 6 and 7), others during summer months (types 2, 3 and 5 ) and one during spring and fall (type 4). By contrast, rain types 8 to 11 seem relatively unseasonal. The seasonality observed in rain type occurrence is in good agreement with the local climatology of rainfall with more snowfall and stratiform rainfalls expected during winter, more sleet showers and rain showers expected in spring and autumn, and more thunderstorm induced convective rainfalls expected during summer. Of course there is no one-to-one match between the rain types identified by our classification and the physical types listed above, for two reasons: first, the rain typing method is purely statistical and we cannot expect a direct match with the physical processes responsible of rainfall generation. All the attempts at linking the rain types with physical processes made in this section should therefore be regarded as qualitative interpretations instead of well-defined relationships. The second reason of the imperfect match is that one rain generation process (e.g. a thunderstorm generating convective rainfalls) can lead to several rain types (e.g. in this case types 2, 3 and 5).

The distributions of the space-time indices displayed in Fig. 9 allow refining the interpretation of the rain types. As an illustration, one rain type typical of each season will be described hereafter in light of the distribution of indices resulting from the classification:

– Rain type 6 is typical of winter months, and probably corresponds to stratiform rainfalls. The rain fields classified in this type are featured by a large fraction of the area of interest covered by rainfall, which leads to a small number of large rain aggregates, and in turn to high connectivity and area indices. Such rain storms are moving eastward and are well correlated in time, which is typical of stratiform rains over Switzerland. Finally the resulting rain intensities are low.

– Rain type 4 is typical of spring and autumn months, and probably corresponds to sleet or rain showers. The corresponding rain fields are very scattered (low fraction of rain coverage and low area index) and are poorly connected. In addition, it

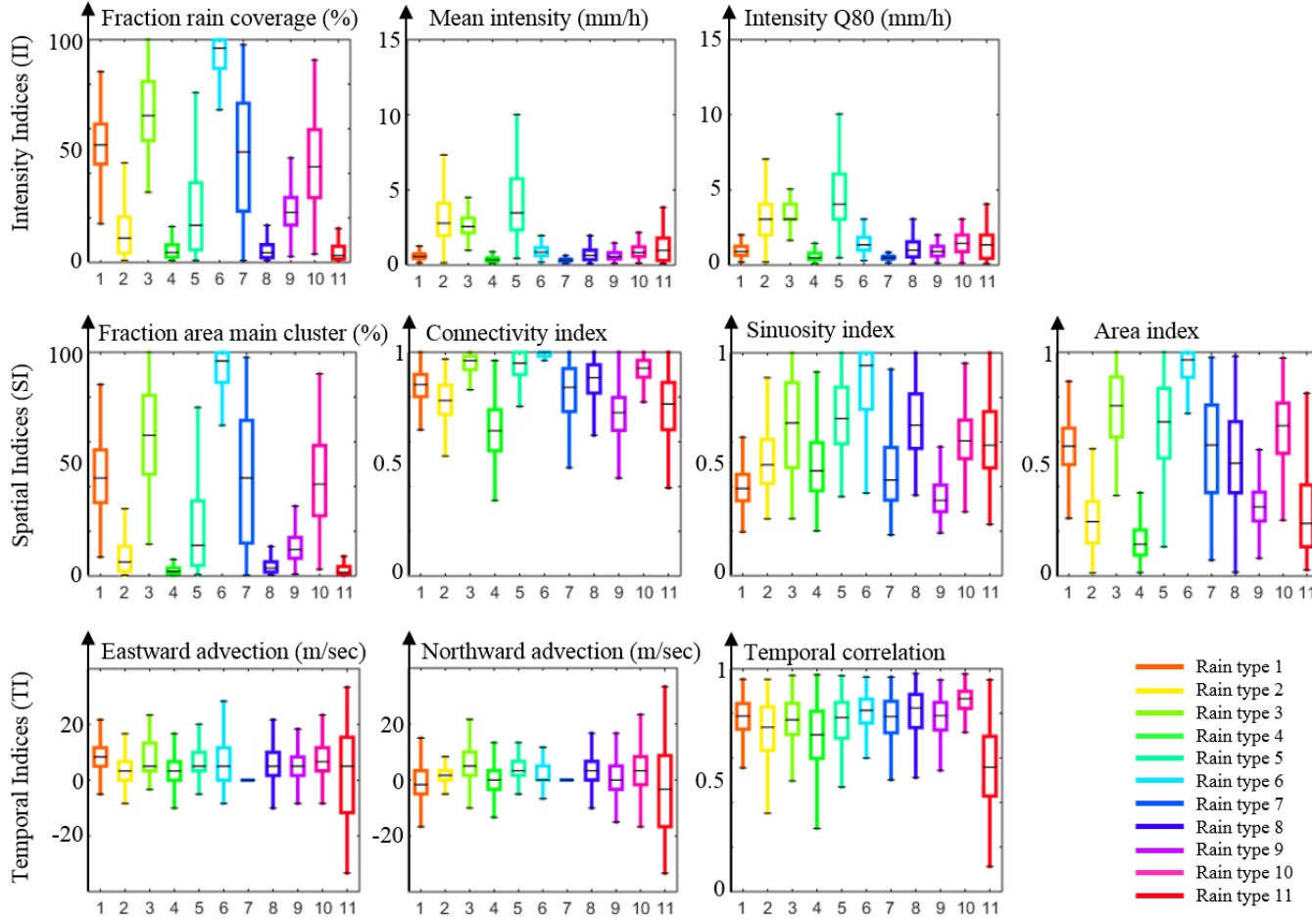

**Figure 9.** Marginal distributions of the indices used for rain type classification. Each graph represents a different index. Colors denote rain types.

is interesting to note that such rain fields have a low temporal correlation, which reflects a strong variability in time as is the case for fast changing mid-season events.

– Rain type 5 is typical of summer months, and probably corresponds to heavy convective rainfalls produced by thunderstorms. Indeed, the rain fields classified in this type generate localized (moderate fraction of rain coverage and contribution of main aggregate) but heavy (high mean and Q80 intensity) rainfalls. Also, such storms mostly originate from the South-West (Eastward and Northward advections > 0) which is typical for large thunderstorms over the Swiss Alps.

It is important to note that even if different rain types dominate at different months, several rain types occur at each month. In other words, at a given location, there is no single typical summer rain or winter rain, but rather a variable collection of rain

types that occur each month. It follows for instance that the rain type 6 is typical of winter months, but it can also occur during summer, although at a lower frequency than the typical summer rain types 2, 3 and 5.

## 4.5 Sensitivity of the rain typing approach to the size of the calibration dataset

To complete the assessment of our rain typing strategy, we study the sensitivity of the classification method to the dataset used to calibrate the GMM model. To this end, the radar images corresponding to the periods covered by the 17 rain events of interest occurring during summer 2017 are typed based on two different GMM models:

– A first GMM model (model A) is calibrated using the radar images of summer 2017 (July and August). This corresponds to the GMM model used in Sect. 4.3, and this classification results in 6 rain types that all occur during the period of interest.

– A second GMM model (model B) is calibrated using the radar images of the full year 2017. This corresponds to the GMM model used in Sect. 4.4. This model results in 11 rain types, 9 of which occur during the period of interest (rain types 7 and 11 are absent).

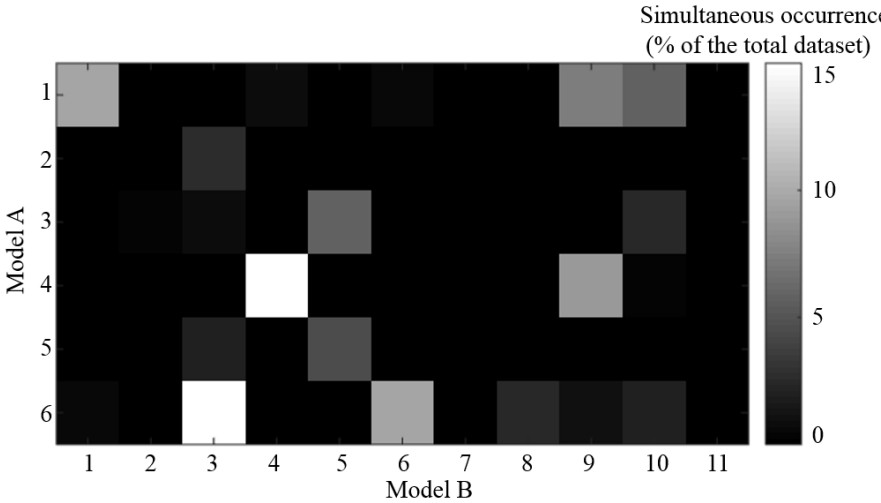

**Figure 10.** Correspondence between rain types derived from two distinct calibration datasets.

Fig. 10 shows how the results of rain typing compare when carried out based on both GMM models A and B. Since the number of rain types differs between the models, there is no one-to-one correspondence. This is because the calibration of model B on a larger dataset allows for a more detailed segmentation of the indices into rain types. Indeed, model A focuses only on the summer, and if some typical winter rain types occasionally occur during this period (which is the case as shown in Sect. 4.3), these rain fields are too infrequent to generate their own cluster. They are then assigned to the closest Gaussian mixture. In contrast, the same rain fields have proportionally more relatives in model B, and they can therefore be grouped in a

separate rain type. This distinct segmentation of the indices space shows that one cannot unequivocally associate a type derived from model B to a type derived from model A. A rain type in model B can straddle the border between two rain types in model A.

Although imperfect, the links between the two classifications are strong. To assess these links, we first make coincide the two classifications by assigning to each rain type of model B its most common counterpart in model A. In this case, the rain types 1, 2, 3, 4, 5, 6, 8, 9, 10 from model B are paired with respectively rain types 1, 3, 6, 4, 3, 6, 6, 4, 1 in model A. By doing so, we obtain a 73.6 % match between the two classifications. Then, we compare the timing of the intra-event rain type transitions for the two raw classifications (i.e. without the previous pairing). Due to the unequivocal links between the two outputs, model B leads to a more fragmented classification (26 transitions instead of 16 for model A). Despite this, most of the transitions in model A (9 among 16) have a counterpart in model B (i.e. a transition appears in model B during the same hour than the transition in model A). This tends to confirm that the intra-event non-stationarities identified in Sect. 4.3 are well defined, and that their detection is robust.

## 5   Discussion

### 5.1   Model dependence in rain typing

The aim of this work is to develop a rain typing strategy able to identify stationary periods for further stochastic rainfall modelling at a sub-daily resolution, with an emphasis on very high resolution models (up to 1 min resolution). Although the main ideas developed throughout this paper can be applied to many different stochastic rainfall models, the detailed settings of the rain typing strategy must be tuned from case to case in order to be compatible with the targeted rainfall model. Indeed, because the final aim is to identify time periods over which stationary statistics can be inferred, the rain typing method necessary depends on the properties to be modeled and the nature of the model. Therefore, we discuss hereafter how the stochastic rainfall model used in this study has influenced the settings of the rain typing method.

As primary data source, raw radar images have been preferred to combined rain gauge – radar products (Sideris et al., 2014) because their higher temporal resolution (10 min instead of 60 min) is more in agreement with the resolution of the stochastic model used (i.e. 1 min). Radar images acquired at higher temporal resolution (e.g. 5 min) are acknowledged to potentially improve the classification, but such images were unfortunately not available for this study. In counterpart of the high temporal resolution of raw radar products, the observed rain intensities can be impaired by local biases, in particular in mountainous regions as the one considered in the present application. However, we think that our method is not significantly affected by such biases because radar shadow effects due to topography are constant in time. Therefore, they do not affect the temporal variability of the rain indices used for classification.

Once the radar product has been selected, another important choice is the size of the area of interest for which the information will be extracted from. Since the study area is very small (3 x 6 km$^2$), we could in theory have taken a window of the same size to analyze radar images. However, a significantly larger window (60 x 60 km$^2$) has been chosen because the space indices used for classification rely on the count and on the spatial arrangement of rain aggregates, which requires several aggregates

within the window of interest for reliable computations. This implies a window slightly larger than the expected scale of rain aggregates, which ranges from 10 km to 30 km in the present case (Benoit et al., 2018). On the other hand, the window extent is restricted by the wish to limit the variety of rain behaviors existing in that window, thereby preserving as much as possible the stationarity of rainfall in space.

Finally and more importantly, the indices selected for classification must be consistent with the statistics embedded in the stochastic rainfall model. Here 10 indices are needed to characterize the evolution of the main features of rainfall considered by the stochastic model, namely the marginal distribution of rain intensity, the spatial arrangement of rain aggregates, and the advection-diffusion of rain storms. This large number of indices reflects the complexity of the stochastic rainfall model in use. In case of a different model (due to e.g. a single site study, a different resolution or another climatology) the set of indices

could be modified.

## 5.2   Consequences of non-stationarity on sub-daily stochastic rainfall modelling

The succession of rain types identified in Sect. 4.3 and 4.4 has two important implications on how stationarity should be regarded in sub-daily stochastic rainfall models.

Due to the variability observed in rainfall statistics, we believe that it is often incorrect to assume rain stationarity over long
periods, such as months or seasons, for characterizing of rainfall statistics. In the perspective adopted in this paper, the seasonality observed in monthly rain statistics is attributed to a variation of rain type occurrence rather than to a smooth change in rainfall behavior (see Sect. 4.4). As such, it would be incorrect to assume constant model parameters over long periods of time because of the risk to mix distinct rainfall statistics during model calibration. This results in the emergence of artificial rain types whose statistics are an average of the statistics of the rain types that actually occur during the period of interest. In
the extreme case, such artificial rain types may not even correspond to any actual rainfall event. This leads in turn to improper space-time dependencies in the simulated rain fields.

A second striking observation is that rainfall statistics can change drastically within a single rain event. As a result, the hypothesis of rain stationarity along entire rain events can be invalidated in some instances. At least in our dataset, such non-stationary events seem relatively frequent (at least 7 non-stationary rain events out of 17 have been identified in our data). This observation
is not new, since it may lead to temporal asymmetry (or temporal irreversibility) in rain rate time series, which is discussed by Müller et al. (2017). Our framework offers a way to deal with this phenomenon through the identification of stationary periods prior to the stochastic modelling of rain. Then, stochastic modelling is carried out separately for each rain type, and the results are merged afterwards. This allows generating synthetic rain fields presenting a temporal asymmetry, even if the stochastic model itself is only capable of generating symmetric rain fields for a given set of model parameters. The temporal asymmetry
is then carried by the temporal arrangement of rain types within a single rain storm.

## 6 Conclusion

This paper proposes a quantitative method to identify stationary rainfall periods, that is, periods during which a set of 10 statistics representative of rainfall space-time behavior at the local scale remains broadly constant. It is based on a classification of radar images into groups of rain fields sharing a similar statistics when observed at high resolution. For reasons of data availability, we focused our investigation on summer rains over the Vaud Alps, Switzerland. However most of the results obtained in this context are expected to be transferable to other seasons, as illustrated in Sect. 4.4, as well as to other mid-latitude areas in cases where extratropical rain storms significantly influence the precipitation regime.

The application of the proposed rain typing method in the context of sub-daily stochastic rainfall modelling shows that our method is able to (1) identify abrupt changes in rainfall statistics during a controlled synthetic experiment, and to (2) delineate relevant stationary periods when applied to an actual dataset. The succession of rain types obtained for our observation dataset is characterized by the coexistence of several distinct rain types, with switches between types occurring within single rain storms. In the context of sub-daily stochastic rainfall modelling, this observation highlights the need of delineating stationary periods based on actual observations rather than subjective assumptions about the rain process.

A possible future work would be to use the proposed method to type simultaneously precipitation fields observed by radars and simulated by numerical models. This may provide a new metric to assess the precipitation component of high-resolution numerical weather or climate models. Indeed, the proper reproduction of rain types and rain type successions in model outputs would indicate a correct simulation of the overall space-time behavior of rainfall by the model.

It could also be interesting to apply the proposed rain typing method to long term archives of radar images in order to investigate the temporal behavior of rain type occurrence. Resulting information could be used as the starting point for the design of a statistical model of rain type occurrence, and in turn a stochastic rain type generator. Coupled with already existing high-resolution stochastic rainfall models, this would allow designing high-resolution stochastic rainfall generators able to reproduce local rainfall statistics over long simulation periods under the assumption of a steady climate. An alternative to the development of a stationary rain type generator would be the design of a synoptically conditioned stochastic rainfall generator (Bárdossy and Plate, 1992; Peleg and Morin, 2014) by linking the occurrence of rain types with the state of the atmosphere simulated by one or several climate models. This could be achieved by assessing the statistical relationships between rain types derived from radar observations and meteorological variables derived from climate model reanalyzes. If strong dependences are found, then the future evolution of climate variables simulated by General Circulation Models (GCMs) could provide precious insights on the possible evolution of rain type occurrence in a changing climate.

*Competing interests.* The authors declare that they have no conflict of interests.

*Acknowledgements.* The source code used for this study is freely available on the following repository: https://github.com/LionelBenoit/Rain_typing.git.

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
