# Peer review of "Dealing with non-stationarity in sub-daily stochastic rainfall models"

_Hydrology and Earth System Sciences, 2018_

## Referee Comment (RC1) · N Peleg (Referee) · 12 Jun 2018

In their paper, the Authors discussed non-stationarity of rainfall in time (e.g. differentiating between rainfall types in time), suggested an approach to automatically divide the rainfall into different types and discussed the importance of extracting the rainfall statistics by their type when applying the statistics in sub-daily stochastic rainfall models. The question of the need to account for non-stationarity in sub-daily stochastic rainfall models is interesting and relevant for the readers of HESS. My comments are mostly minor, except for one major issue that I encourage the Authors to address: in the case study the Authors present their results based on only 10 realizations. Likely, more than 10 realizations are needed in order to quantify the natural (stochastic) variability in rainfall and to statistically identify the mean signal of the rainfall. For example, looking at Figure 6, it is quite clear that with 10 realizations the variability in rainfall is quite high and the discussion of the results (i.e. the differences in cumulative rainfall amounts between rainfall types H1, H2 and H3) could be biased to the fact that only part of the natural variability is presented. This issue can be easily resolved with increasing the number of realizations simulated per rainfall type. Other than that, I found the paper interesting, well written and structured. The method suggested by the Authors is sound and I believe the hydrological community, especially the researchers using weather generators, will benefit from the paper. Please find my recommendations for some text editing along with some minor comments to address below.

[page line]

[2 26] ". 'space' is used in this paragraph with two different meanings. First, in relation to the domain ("the question of stationarity in space is not addressed here because we focus on regional to local areas", and then in relation to the structure of rainfall within the domain ("The proposed framework relies on the classification of radar images based on their space-time features"). I found it to be a bit confusing.

[3 18] "…small enough to ensure spatial stationarity". The study area is characterized with a complex terrain, as seen from the figure illustrating the domain. What are the height differences and how do you know for sure that stationarity is preserved in space? To check spatial stationarity one must have a very long (>30 years) observation set. I suggest using one of MeteoSwiss gridded product (e.g. HiresD) which is freely available to check if the assumption of spatial stationarity is valid.

[3 20] The Alps mask and bias rainfall estimates from the weather radar over some areas in Switzerland. How good are the rainfall estimates from the Swiss weather radar system for this region?

[Figure 1] Please also add a scale bar in plot (a). I suggest adding an arrow indicating the advection direction and speed (i.e. using arrow length) in plot (c).

[5 2] "This change in the space-time features is very rapid and takes place in less than 30 min". Very interesting. If the changes are so rapid, why not using the weather radar data from MeteoSwiss at 5-min resolution? Results should be more robust. I guess data at 5-min resolution was not available for this study, but this is a point (of using higher temporal resolution data) you might want to later discuss.

[5 16] "Quantile 80% of rain intensities". Why 80%? Why not 95% or even 99% to explicitly account for extreme rainfall intensities? Please give the reason for choosing the 80th quantile.

[5 18] "…and applied here in the context of rain fields occurring at mid-latitudes under a temperate climate". Does it matter for which climate it is applied for?

[5 19] "They are computed based on binary images representing rain masks". I suggest adding a reference here to Fig. 2.

[5 21] "clusters". It seems that sometimes 'clusters' are used and sometimes 'cells' are used to describe the same rainfall features. Please be consist.

[6 26] "…we adopt an approach based on a Gaussian Mixture Model classifier (GMM)". Why? Using GMM for clustering is one option, but other methods to cluster data are available. Please give the reason to favorite GMM on other methods. Also - I guess the build-in Matlab functions were used?

[8 4-5] A repetitive. Consider deleting.

[8 5] "In addition, it can be desired to avoid noisy successions of rain types". This sentence is not clear to me.

[Table 1] "Type 3" is missing.

[10 8] "…convective rains (Type 1), thunderstorm-related heavy showers (Type 2), and stratiform rains (Type 3)". Why not following the same types you are presenting in Figure 1? I was confused at first reading as I had in mind that the rainfall types presented in Figure 1 are the one that will be discussed and presented throughout the paper. And - I would interpret both types 1 and 2 as convective events, characterized with high intensity rainfall and associated with thunderstorm, but one is more correlated in space than the other. I found the names given to type 1 and 2 a bit misleading.

[11 2] "synthetic images". At 1-km and 10-min, right?

[11 3] "Results show that the proposed method can consistently detect the prescribed rain types and their temporal evolution, for all realizations". Please support this statement with some numbers. How good are the results? Looking at the figure, it seems close to 100% fit.

[12 2] "10 realizations". I recommend increasing the number of realizations (to 30, or even 50) to better account for the natural (stochastic) variability of climate. Examining the results in Figure 6 - the signal of under/overestimating will be clearer with more realizations plotted.

[Discussion and conclusion]. I am missing some discussion on the advantages and limitations of using GGM comparing to other cluster methods, the sensitivity of the results depending on the threshold (10% and 60 min) that you were using, discussing if the 6 rainfall types that were automatically chosen are really differ (physically and statistically) from each other - or if some rainfall types could have been consider as the same one to reduce the number of types, etc. The discussion part should be "thicken" and should be separated from the conclusion part.

[15 21-25] In this context, I would like to mention three of our previous studies (Peleg and Morin, 2012, 2014; Peleg et al., 2015) in which we follow the framework suggested here, i.e. we classified rainfall types based on climate variables taken from re-analysis data (rainfall statistics are therefore based on rainfall images from a weather radar that are linked to different synoptic conditions/systems), used the statistics to train a sub-hourly distributed rainfall generator to simulate rainfall at high-resolution (5-min and 250-m) for both present and future climates (linked to GCMs). I believe that some parts of the analysis presented in these papers (e.g. cluster analysis of rainfall types, extracting space-time statistics from radar images to train rainfall generators) is also relevant here.

Peleg, N., Morin, E., (2012) Convective rain cells: Radar-derived spatio-temporal characteristics and synoptic patterns over the Eastern Mediterranean. Journal of Geophysical Research, 117,D15116.

Peleg, N., Morin, E., (2014) Stochastic convective rain-field simulation using a high-resolution synoptically conditioned weather generator (HiReS-WG). Water Resources Research, 50(3), 2124-2139.

Peleg, N., Shamir, E., Georgakakos, KP., Morin, E., (2015) A framework for assessing hydrological regime sensitivity to climate change in a convective rainfall environment: A case study of two medium-sized eastern Mediterranean catchments, Israel. Hydrology and Earth System Sciences, 19, 567-581.

---

## Referee Comment (RC2) · Anonymous Referee #2 · 2 Jul 2018

This article suggests a novel approach of rainfall typing that can be applied on a series of radar rainfall (or any pixlelated) data.

My major concern on the suggested approach is as follow:

About the main idea:

1. The authors suggests that a rainfall type can alter from convective to frontal, or the other way around. I suspect whether this assumption can be verified (did you find any supporting literature?). I do not think that the similar conclusion would have been drawn if the authors adopted a Lagrangian approach instead of the Eulerian approach of fixing spatial window.

About the classification model:

[Figure]

2. Are the 10 rainfall characteristics used to parameterize each rainfall imagery independent? I suspect validity of the 10-dimensional Gaussian mixture model because most Copula tend to show awkward behavior as its dimension exceeds 2. Would you show a way to validate the GMM model you developed?

3. I do not believe that the distribution of all 10 variables used for GMM classification has a normal distribution, which is a fundamental model assumption. For example, most rainfall intensity in a time series (in your case, it corresponds to the II-2) has a skewed distribution. You may want to use a type of transfer function to convert the original variables to be normally distributed, and then run the GMM classification algorithm.

4. If you cannot validate the previous two points, I suggest you perform the principal component analysis to extract the principle variables, and then apply a simpler clustering algorithms based on the Euclidean distance (e.g. K-means clustering, Hierarchical clustering).

About the method of validation:

5. I believe that the result will be much more stronger if the analysis is performed on the years of the radar rainfall data at multiple locations. I agree with the view of the authors that the absolute validation of the result is not possible because we cannot see the nature thoroughly. However, it cannot be an excuse of not validating your model for a variety of situations.

6. In this view, running a sensitivity analysis will be more helpful. For example, you can change the size of the spatial window, or run the model at different locations with different meteorology, and see how your model behaves.

———————————————

---

## Author Comment (AC1) · 26 Jul 2018

Dear Editor and Reviewers,

Thank you for your detailed comments and suggestions about our manuscript entitled "Dealing with non-stationarity in sub-daily stochastic rainfall models".

To capitalize on your propositions of improvement, we suggest to thicken the validation and discussion parts of our manuscript. To this end, we consider to modify the plan of the paper as detailed hereafter. The content of the new sections is introduced together with our point-by-point responses to the comments of the reviewers.

Hoping that our responses answer your concerns, and that our propositions of improvements will fulfil your expectations,

Best regards,

Lionel Benoit, Mathieu Vrac and Gregoire Mariethoz.

General overview of the proposed changes:

Considering the comments and recommendations of the reviewers, we plan to add some material to the validation and discussion parts of our paper. The guidelines of the paper will be amended as follow:

1. Introduction
2. Overview of rainfall space-time patterns observed in radar images
3. Assessing rain statistics stationarity from radar images
    3.1. Extracting space-time information from radar images
    3.2. Classification of radar images based on rainfall space-time statistics
4. Validation and application
    4.1. Stochastic rainfall model
    4.2. Detection of rainfall non-stationarity in a controlled setting
    4.3. Impact of rainfall non-stationarity on stochastic modelling of an actual dataset
    4.4. Seasonality of rain type occurrence [new material]
    4.5. Sensitivity of the rain typing approach to the size of the calibration dataset [new material]
5. Discussion [now separated from the conclusion]
    5.1. Model dependence of rain typing [new material]
    5.2. Consequences of non-stationarity on sub-daily stochastic rainfall modelling [new material + part of the current conclusion section]
6. Conclusion

The new sections will be written and the others amended to answer the concerns of the referees as detailed hereafter. In the following RC denotes a reviewer comment and AR denotes our response to the comment.
* * *
Responses to the comments of Reviewer #1:

RC: My comments are mostly minor, except for one major issue that I encourage the Authors to address: in the case study the Authors present their results based on only 10 realizations. Likely, more

than 10 realizations are needed in order to quantify the natural (stochastic) variability in rainfall and to statistically identify the mean signal of the rainfall.

AR: We agree with this comment. In the revised version of the paper, we will assess our method based on a set of 50 realizations instead of 10 in the current version.

RC: [2 26] ". 'space' is used in this paragraph with two different meanings. First, in relation to the domain ("the question of stationarity in space is not addressed here because we focus on regional to local areas", and then in relation to the structure of rainfall within the domain ("The proposed framework relies on the classification of radar images based on their space-time features"). I found it to be a bit confusing.

AR: We agree with this comment. We will modify our manuscript accordingly.

RC: [3 18] "…small enough to ensure spatial stationarity". The study area is characterized with a complex terrain, as seen from the figure illustrating the domain. What are the height differences and how do you know for sure that stationarity is preserved in space? To check spatial stationarity one must have a very long (>30 years) observation set. I suggest using one of MeteoSwiss gridded product (e.g. HiresD) which is freely available to check if the assumption of spatial stationarity is valid.

AR: We agree that spatial stationarity is an important question, which is linked to the nature of the terrain, but also to the size of the area of interest. However, in the present study, we choose to focus on the temporal non-stationarity of rainfall statistics. We therefore consider the size of the target area as a setting of our method, and the stationarity of rainfall within this area as a prerequisite modelling assumption. The choice of the size of the target area (and the related question of spatial non-stationarity) therefore depends on the targeted application and, in turn, it depends on the stochastic rainfall model that is being applied (and for which we are looking for stationary periods). That is why this question of the size of the study window and the stationarity of rainfall within the area of interest will be discussed in details in the new section "5.1 Model dependence of rain typing" in light of the specific stochastic rainfall model used in the present study.

RC: [3 20] The Alps mask and bias rainfall estimates from the weather radar over some areas in Switzerland. How good are the rainfall estimates from the Swiss weather radar system for this region?

AR: We agree that the quality of available radar data is uneven depending of the area of interest, and therefore depending on the problem being addressed. Therefore, this issue will also be discussed in the new section "5.1 Model dependence of rain typing."

RC: [Figure 1] Please also add a scale bar in plot (a). I suggest adding an arrow indicating the advection direction and speed (i.e. using arrow length) in plot (c).

AR: We agree with this comment. We will modify our manuscript accordingly.

RC: [5 2] "This change in the space-time features is very rapid and takes place in less than 30 min". Very interesting. If the changes are so rapid, why not using the weather radar data from MeteoSwiss at 5-min resolution? Results should be more robust. I guess data at 5-min resolution was not available for this study, but this is a point (of using higher temporal resolution data) you might want to later discuss.

AR: Exactly, we used 10-min resolution data because 5-min resolution data were not available to us. We will add a sentence in the new version of our manuscript to make this point more clear.

RC: [5 16] "Quantile 80% of rain intensities". Why 80%? Why not 95% or even 99% to explicitly account for extreme rainfall intensities? Please give the reason for choosing the 80th quantile.

AR: The idea is to characterize high rain intensities, but the value of 80% is arbitrary. Note that estimating a robust 99% quantile requires an amount of data that we do not have necessarily in one single radar image cropped over a 60 x 60km$^2$ area. We will add a comment in the manuscript to precise it.

RC: [5 18] "…and applied here in the context of rain fields occurring at mid-latitudes under a temperate climate". Does it matter for which climate it is applied for?

AR: You are right, it does not matter at this point. The climate of the study area matters to interpret the rain types that are obtained and their frequency of occurrence (cf new section 4.4. and conclusion), but not for the selection of the space indices. We will delete this sentence from 'and applied here…'.

RC: [5 19] "They are computed based on binary images representing rain masks". I suggest adding a reference here to Fig. 2.

AR: We agree with this comment. We will modify our manuscript accordingly.

RC: [5 21] "clusters". It seems that sometimes 'clusters' are used and sometimes 'cells' are used to describe the same rainfall features. Please be consist.

AR: We agree with this comment. We will modify our manuscript accordingly.

RC: [6 26] "…we adopt an approach based on a Gaussian Mixture Model classifier (GMM)". Why? Using GMM for clustering is one option, but other methods to cluster data are available. Please give the reason to favorite GMM on other methods. Also - I guess the build-in Matlab functions were used?

AR: We adopted the GMM classifier because this framework allows for a consistent automatic selection of the number of clusters through a model selection approach based on the BIC criterion. In addition, this framework does not require any assumption about the joint distribution of the data used for classification, which is useful in the present case since we do not have prior information about the distribution of the rain indices. We will add a sentence to further justify our choice in the new version of the paper.
Regarding the implementation, we used build-in Matlab functions fitgmdist and cluster. We forgot to mention that. We will specify it in the next version.

RC: [8 4-5] A repetitive. Consider deleting.

AR: We agree with this comment. We will modify our manuscript accordingly.

RC: [8 5] "In addition, it can be desired to avoid noisy successions of rain types". This sentence is not clear to me.

AR: We agree with this comment. We will modify our manuscript accordingly.

RC: [Table 1] "Type 3" is missing.

AR: We agree with this comment. We will modify our manuscript accordingly.

RC: [10 8] "…convective rains (Type 1), thunderstorm-related heavy showers (Type 2), and stratiform rains (Type 3)". Why not following the same types you are presenting in Figure 1? I was confused at first reading as I had in mind that the rainfall types presented in Figure 1 are the one that will be discussed and presented throughout the paper. And - I would interpret both types 1 and 2 as convective events, characterized with high intensity rainfall and associated with thunderstorm, but one is more correlated in space than the other. I found the names given to type 1 and 2 a bit misleading.

AR: To avoid confusion, in the revised version of our manuscript, we will try to identify rain types by numbers only, and not using names that refer to physical properties. The reference to the nature of the rain generation processes (e.g. convective, thunderstorm , stratiform, etc.) will be confined to the introductive sections 1. and 2., as well as to the discussion of the physical meaning of the rain types identified in new section "4.4 Seasonality of rain type occurrence".

RC: [11 2] "synthetic images". At 1-km and 10-min, right?

AR: Yes, you are right.

RC: [11 3] "Results show that the proposed method can consistently detect the prescribed rain types and their temporal evolution, for all realizations". Please support this statement with some numbers. How good are the results? Looking at the figure, it seems close to 100% fit.

AR: We agree with this comment. We will modify our manuscript accordingly.

RC: [12 2] "10 realizations". I recommend increasing the number of realizations (to 30, or even 50) to better account for the natural (stochastic) variability of climate. Examining the results in Figure 6 - the signal of under/overestimating will be clearer with more realizations plotted.

AR: You are right, we will perform 50 realizations instead of 10 to assess our method.

RC: I am missing some discussion on […] the sensitivity of the results depending on the threshold (10% and 60 min) that you were using.

AR: This will be discussed in the new section: "5.1 Model dependence of rain typing".

RC: I am missing some discussion on […] discussing if the 6 rainfall types that were automatically chosen are really differ (physically and statistically) from each other - or if some rainfall types could have been consider as the same one to reduce the number of types, etc.

AR: This will be discussed on an annual basis in the new section "4.4 Seasonality of rain type occurrence".

RC: The discussion part should be "thicken" and should be separated from the conclusion part.

AR: Agree. The discussion will be a section by itself, and it will be thicken by the addition of a new sub-section "5.1 Model dependence of rain typing". The second part of the discussion, entitled "5.2 Consequences of non-stationarity on sub-daily stochastic rainfall modelling" will encompass the discussion part of the current "Discussion and conclusion" section as well as additional comments about the impact of the observed rainfall non-stationarity on stochastic rainfall experiments.

RC: [15 21-25] In this context, I would like to mention three of our previous studies (Peleg and Morin, 2012, 2014; Peleg et al., 2015) in which we follow the framework suggested here, i.e. we classified rainfall types based on climate variables taken from re-analysis data (rainfall statistics are therefore based on rainfall images from a weather radar that are linked to different synoptic

conditions/systems), used the statistics to train a sub-hourly distributed rainfall generator to simulate rainfall at high-resolution (5-min and 250-m) for both present and future climates (linked to GCMs). I believe that some parts of the analysis presented in these papers (e.g. cluster analysis of rainfall types, extracting space-time statistics from radar images to train rainfall generators) is also relevant here.

AR: Thank you for this comment and for sharing your previous studies. We will further discuss the future work part in comparison with the papers you mention.

---

## Author Comment (AC2) · 26 Jul 2018

Dear Editor and Reviewers,

Thank you for your detailed comments and suggestions about our manuscript entitled "Dealing with non-stationarity in sub-daily stochastic rainfall models".

To capitalize on your propositions of improvement, we suggest to thicken the validation and discussion parts of our manuscript. To this end, we consider to modify the plan of the paper as detailed hereafter. The content of the new sections is introduced together with our point-by-point responses to the comments of the reviewers.

Hoping that our responses answer your concerns, and that our propositions of improvements will fulfil your expectations,

Best regards,

Lionel Benoit, Mathieu Vrac and Gregoire Mariethoz.

General overview of the proposed changes:

Considering the comments and recommendations of the reviewers, we plan to add some material to the validation and discussion parts of our paper. The guidelines of the paper will be amended as follow:

1. Introduction
2. Overview of rainfall space-time patterns observed in radar images
3. Assessing rain statistics stationarity from radar images
   3.1. Extracting space-time information from radar images
   3.2. Classification of radar images based on rainfall space-time statistics
4. Validation and application
   4.1. Stochastic rainfall model
   4.2. Detection of rainfall non-stationarity in a controlled setting
   4.3. Impact of rainfall non-stationarity on stochastic modelling of an actual dataset
   4.4. Seasonality of rain type occurrence [new material]
   4.5. Sensitivity of the rain typing approach to the size of the calibration dataset [new material]
5. Discussion [now separated from the conclusion]
   5.1. Model dependence of rain typing [new material]
   5.2. Consequences of non-stationarity on sub-daily stochastic rainfall modelling [new material + part of the current conclusion section]
6. Conclusion

The new sections will be written and the others amended to answer the concerns of the referees as detailed hereafter. In the following RC denotes a reviewer comment and AR denotes our response to the comment.
* * *
Responses to the comments of Reviewer #2:

RC: The authors suggests that a rainfall type can alter from convective to frontal, or the other way around. I suspect whether this assumption can be verified (did you find any supporting literature?). I

do not think that the similar conclusion would have been drawn if the authors adopted a Lagrangian approach instead of the Eulerian approach of fixing spatial window.

AR: You are pointing out the critical choice of the reference that is used to perform the rain typing.
As you mention, in our approach, we adopt an Eulerian reference, i.e. a reference that is fixed with regard to the Earth surface. We think that it is coherent with the target of assessing the temporal stationarity of rainfall statistics at a given location (or over a restricted area) for the purpose of stochastic rainfall modelling.
Because of the choice of this Eulerian reference, the rain type that is 'flying over' the area of interest can change along time due to the travel of distinct storms over the area. This is this succession of rain types that we are trying to identify. This is in line with a large literature about rainfall typing (e.g. Biggerstaff and Listemaa, 2000; Llasat, 2001), with the difference that the number of rain types considered is usually lower, and that the rain types are usually defined from physical properties (e.g. convective vs stratiform rains) while in this study we characterize rain types from their statistical properties.
Therefore our conclusion of sharp transitions between rain types (in an Eulerian reference) does not implies that the nature of a given 'piece of clouds' defined in a Lagrangian reference (i.e. moving with the rain storm) will change along time, but only that distinct 'sub-storms' with distinct features follow each other over the area of interest under the influence of storm advection.
We acknowledge that the question of changing rain types in a Lagrangian reference frame is an interesting and open question, but it is thought to be out of the scope of the present paper. Moreover we do not know any supporting literature about this point, and we do not aim to show that this phenomenon actually occurs. This would require homogenized radar data over large areas (in any case larger than Switzerland) in order to follow rain storms to track potential changes of their statistics, and for now we do not have access to such data.
In order to dispel any doubts about the aim of our work, we will improve the sections "1. Introduction" and "2. Overview of rainfall space-time patterns observed in radar images" by introducing the notions of Eulerian and Lagrangian references and by clearly specifying that an Eulerian approach has been followed. In addition we will better contextualize our study in comparison with former works about rain typing, including the references proposed by Reviewer #1.

RC: Are the 10 rainfall characteristics used to parameterize each rainfall imagery independent?

AR: No, we do not assume that the 10 indices used to characterize the rain fields are independent. That is why we use a full covariance model (i.e., accounting for correlations) for the covariance matrices implied in the GMM model, following the guidelines formulated in the review paper by Fraley and Raftery (2002) about GMM clustering.

RC: I suspect validity of the 10-dimensional Gaussian mixture model because most Copula tend to show awkward behavior as its dimension exceeds 2. Would you show a way to validate the GMM model you developed?

AR: GMM clustering has already been successfully applied to relatively high dimensional problems (more than 10 components). For instance Vrac & Yiou (2010), Rust et al. (2010) or Pernin et al. (2016) applied GMM clustering to problems of dimension 10 to 20.
Regarding the implementation of the GMM clustering, we used the build-in Matlab functions fitgmdist and cluster. We forgot to mention it in the first version of the manuscript, but we will add this information in the next version.

RC: I do not believe that the distribution of all 10 variables used for GMM classification has a normal distribution, which is a fundamental model assumption. For example, most rainfall intensity in a time series (in your case, it corresponds to the II-2) has a skewed distribution. You may want to use a type

of transfer function to convert the original variables to be normally distributed, and then run the GMM classification algorithm.

AR: The GMM classification does not require normal distributions. Indeed, the idea behind GMM classification is that any multivariate distribution can be approximated by a mixture of multivariate Gaussian distributions if enough components are added. For more insights about GMM classification we refer to the review paper by Fraley and Raftery (2002).
We believe that this misunderstanding originates from our too brief description of the GMM classifier. Therefore, in the next version of our manuscript, we will expend the description of this method.

RC: If you cannot validate the previous two points, I suggest you perform the principal component analysis to extract the principle variables, and then apply a simpler clustering algorithms based on the Euclidean distance (e.g. K-means clustering, Hierarchical clustering).

AR: We hope that our responses to the two previous points convinced you that GMM is a rational choice to carry the classification in the specific context of our study. In addition, we would like to mention (cf response to Reviewer #1 comments) that we preferred GMM to other classification methods because this framework allows for a consistent automatic selection of the number of clusters through a model selection approach based on the BIC criterion.

RC: I believe that the result will be much more stronger if the analysis is performed on the years of the radar rainfall data at multiple locations. I agree with the view of the authors that the absolute validation of the result is not possible because we cannot see the nature thoroughly. However, it cannot be an excuse of not validating your model for a variety of situations.

AR: We agree that for a more exhaustive assessment, our framework should be applied to a wider variety of situations. However, since our method focuses on temporal non-stationarities, we prefer to focus on different seasons rather than on different locations. We therefore plan to develop an assessment of our method on a variety of situations in the new section "4.4 Seasonality of rain type occurrence". In this section we will apply our rain typing approach to an entire year of data (2017) in order to qualitatively assess the seasonality of rain type occurrence inferred by our classification method. Therefore, multiple meteorological situations will be investigated including e.g. stratiform rain events, convective thunderstorms, spring showers, etc.

RC: In this view, running a sensitivity analysis will be more helpful. For example, you can change the size of the spatial window, or run the model at different locations with different meteorology, and see how your model behaves.

AR: We agree with the need of a sensitivity analysis to further validate our method, but here again we prefer to focus on the time axis. We therefore plan to investigate the sensitivity of our method to the length of the calibration dataset in a new section: "4.5 Sensitivity of the rain typing approach to the size of the calibration dataset". In this section, we will split the rainy periods of the summer of 2017 into stationary rain events based on two calibrations of the GMM classifier: using the data of the summer of 2017 only, and using the data of the whole year 2017. The two classifications will be compared and we will discuss the robustness of the proposed approach to the choice of the modelling domain (in time).

References:

Biggerstaff, M. I. and S. A. Listemaa (2000). "An Improved Scheme for Convective/Stratiform Echo Classification Using Radar Reflectivity." Journal of applied meteorology **39**: 2129-2150.

Fraley, C. and A. E. Raftery (2002). "Model-Based Clustering, Discriminant Analysis, and Density Estimation." Journal of the American Statistical Association **97**(458): 611-631.

Llasat, M. C. (2001). "An objective classification of rainfall events on the basis of their convective features: application to rainfall intensity in the northeast of spain." International Journal of Climatology **21**: 1385-1400.

Pernin, J., Vrac, M., M., Crevoisier, C., Chédin, A. (2016) Mixture model-based air mass classification: A probabilistic view of thermodynamic profiles. Adv. Stat. Clim. Meteorol. Oceanogr., 2, 115–136, doi:10.5194/ascmo−2−115−2016

Rust, H., Vrac, M., Lengaigne, M., Sultan, B. (2010) Quantifying differences in circulation patterns based on probabilistic models: IPCC-AR4 multi-model comparison for the North Atlantic. Journal of Climate, 23, 6573-6589, doi: 10.1175/2010JCLI3432.1

Vrac, M. and Yiou, P. (2010) Weather regimes designed for local precipitation modelling: Application to the Mediterranean basin. J. Geophys. Res. - Atmospheres, 115, D12103, doi:10.1029/2009JD012871

---

## Author Response (AR1)

Dear Editor and Reviewers,

Thank you for your detailed comments and suggestions about our manuscript entitled "Dealing with non-stationarity in sub-daily stochastic rainfall models".

The paper has been revised accordingly. Please find hereafter the details of the changes and an item-by-item response (in green) to your comments (in black). If our corrections are direct implementations of your remarks, the answer to the comment is simply 'Ok'. Please note that the page and line numbers indicated hereafter refer to the ones of the revised manuscript.

Hoping that these modifications will fulfill your expectations,

Best regards,

Lionel Benoit, Mathieu Vrac and Gregoire Mariethoz.
* * *
Responses to the comments of Reviewer#1:

RC: My comments are mostly minor, except for one major issue that I encourage the Authors to address: in the case study the Authors present their results based on only 10 realizations. Likely, more than 10 realizations are needed in order to quantify the natural (stochastic) variability in rainfall and to statistically identify the mean signal of the rainfall.

AR: In the revised version of the paper, our method is now assessed based on a set of 50 realizations (p 11, l 32). The conclusions about the performance of the different hypotheses do not change much, except that H3 reproduces a bit less well the temporal structure of the rain compared to previous tests based on 10 realizations only. Therefore, the discussion about the performance of H3 has been rewritten accordingly (p13, l 7-11).

RC: [2 26] 'space' is used in this paragraph with two different meanings. First, in relation to the domain ("the question of stationarity in space is not addressed here because we focus on regional to local areas", and then in relation to the structure of rainfall within the domain ("The proposed framework relies on the classification of radar images based on their space-time features"). I found it to be a bit confusing.

AR: This paragraph has been rewritten to avoid confusion between the two meanings of the word 'space' (p 2, l 27-33). In particular, we now explain in brackets what we exactly mean with "stationarity in space", i.e., "constant statistics over the whole area of interest".

RC: [3 18] "…small enough to ensure spatial stationarity". The study area is characterized with a complex terrain, as seen from the figure illustrating the domain. What are the height differences and how do you know for sure that stationarity is preserved in space? To check spatial stationarity one must have a very long (>30 years) observation set. I suggest using one of MeteoSwiss gridded product (e.g. HiresD) which is freely available to check if the assumption of spatial stationarity is valid.

AR: The stationarity in space is now regarded as a prerequisite modelling assumption as mentioned in the introduction (p 2, l 29). The mention to "small enough to ensure spatial stationarity" in section 2 has thus been deleted. To replace the deleted paragraph, the notion of stationarity in space is now

discussed in section 5.1 (p 18, l 30-34; p19, l 1-4) in relation to the stochastic rainfall model used in the present study.

RC: [3 20] The Alps mask and bias rainfall estimates from the weather radar over some areas in Switzerland. How good are the rainfall estimates from the Swiss weather radar system for this region?

AR: The quality of the radar images used in this study, and in particular the possible biases due to mountain masks is now discussed in section 5.1 (p 18, l 22-29).

RC: [Figure 1] Please also add a scale bar in plot (a). I suggest adding an arrow indicating the advection direction and speed (i.e. using arrow length) in plot (c).

AR: Ok.

RC: [5 2] "This change in the space-time features is very rapid and takes place in less than 30 min". Very interesting. If the changes are so rapid, why not using the weather radar data from MeteoSwiss at 5-min resolution? Results should be more robust. I guess data at 5-min resolution was not available for this study, but this is a point (of using higher temporal resolution data) you might want to later discuss.

AR: The temporal resolution of the radar images used in this study as well as the interest of using higher temporal resolution data (e.g. an image every 5min) are now discussed in section 5.1 (p 18, l 22-29).

RC: [5 16] "Quantile 80% of rain intensities". Why 80%? Why not 95% or even 99% to explicitly account for extreme rainfall intensities? Please give the reason for choosing the 80th quantile.

AR: The idea is to characterize high rain intensities, but the value of 80% is arbitrary. Note that estimating a robust 99% quantile requires an amount of data that we do not have necessarily in one single radar image cropped over a 60 x 60km² area. We added a sentence (p 5, l 15) to clearly specify that this index aims at characterizing heavy rainfalls.

RC: [5 18] "...and applied here in the context of rain fields occurring at mid-latitudes under a temperate climate". Does it matter for which climate it is applied for?

AR: We deleted this sentence from 'and applied here...' (p 5, l 17).

RC: [5 19] "They are computed based on binary images representing rain masks". I suggest adding a reference here to Fig. 2.

AR: Ok.

RC: [5 21] "clusters". It seems that sometimes 'clusters' are used and sometimes 'cells' are used to describe the same rainfall features. Please be consist.

AR: We replaced 'clusters' by 'aggregates' to avoid confusion with clusters in the classification, and in addition we removed the use of 'cells' when referring to 'aggregates'.

RC: [6 26] "...we adopt an approach based on a Gaussian Mixture Model classifier (GMM)". Why? Using GMM for clustering is one option, but other methods to cluster data are available. Please give the reason to favorite GMM on other methods. Also - I guess the build-in Matlab functions were used?

AR: In the revised version of our manuscript, we specify that we use the GMM classifier because it allows for an automatic selection of the number of clusters, and because it works properly for virtually all possible joint distributions of indices (p 6, l 22-24). In addition, we mention which Matlab functions are used for the processing (p 7, l 6 and 11).

RC: [8 4-5] A repetitive. Consider deleting.

AR: Ok.

RC: [8 5] "In addition, it can be desired to avoid noisy successions of rain types". This sentence is not clear to me.

AR: This part has been rewritten for improved clarity.

RC: [Table 1] "Type 3" is missing.

AR: Type 3 has now been added in Table 1.

RC: [10 8] "…convective rains (Type 1), thunderstorm-related heavy showers (Type 2), and stratiform rains (Type 3)". Why not following the same types you are presenting in Figure 1? I was confused at first reading as I had in mind that the rainfall types presented in Figure 1 are the one that will be discussed and presented throughout the paper. And - I would interpret both types 1 and 2 as convective events, characterized with high intensity rainfall and associated with thunderstorm, but one is more correlated in space than the other. I found the names given to type 1 and 2 a bit misleading.

AR: In the revised version of the manuscript, we avoid using names that refer to physical properties of rainfall, except in section 4.4. We hope that this change will avoid misinterpretations of the rain types we identify.

RC: [11 2] "synthetic images". At 1-km and 10-min, right?

AR: Ok. We added a sentence to clarify it (p 9, l 28).

RC: [11 3] "Results show that the proposed method can consistently detect the prescribed rain types and their temporal evolution, for all realizations". Please support this statement with some numbers. How good are the results? Looking at the figure, it seems close to 100% fit.

AR: This statement is now supported by numbers (p 11, l 1-2), and indeed the fit is between 92.6% and 97%.

RC: [12 2] "10 realizations". I recommend increasing the number of realizations (to 30, or even 50) to better account for the natural (stochastic) variability of climate. Examining the results in Figure 6 - the signal of under/overestimating will be clearer with more realizations plotted.

AR: The number of realizations has been increased to 50 (p 11, l 32).

RC: I am missing some discussion on […] the sensitivity of the results depending on the threshold (10% and 60 min) that you were using.

AR: The sensitivity of the results depending on the setting of the method and on the stochastic model in use is now discussed in section 5.1 (p 18, l 15-33; p 19, l 1-10).

RC: I am missing some discussion on […] discussing if the 6 rainfall types that were automatically chosen are really differ (physically and statistically) from each other - or if some rainfall types could have been consider as the same one to reduce the number of types, etc.

AR: The rain types identified by our method and their physical meaning are now discussed in details in section 4.4 (from p 13, l 15 to p 17, l 2).

RC: The discussion part should be "thicken" and should be separated from the conclusion part.

AR: The discussion is now a section by itself (section 5), and has been significantly thickened.

RC: [15 21-25] In this context, I would like to mention three of our previous studies (Peleg and Morin, 2012, 2014; Peleg et al., 2015) in which we follow the framework suggested here, i.e. we classified rainfall types based on climate variables taken from re-analysis data (rainfall statistics are therefore based on rainfall images from a weather radar that are linked to different synoptic conditions/systems), used the statistics to train a sub-hourly distributed rainfall generator to simulate rainfall at high-resolution (5-min and 250-m) for both present and future climates (linked to GCMs). I believe that some parts of the analysis presented in these papers (e.g. cluster analysis of rainfall types, extracting space-time statistics from radar images to train rainfall generators) is also relevant here.

AR: In the revised version of our manuscript we contextualize the future work in relation to synoptically conditioned stochastic rainfall generators and cite one of the papers proposed by Reviewer#1 (see the last paragraph of our conclusion, p 20, l 18-28).

Responses to the comments of Reviewer#2:

RC: The authors suggests that a rainfall type can alter from convective to frontal, or the other way around. I suspect whether this assumption can be verified (did you find any supporting literature?). I do not think that the similar conclusion would have been drawn if the authors adopted a Lagrangian approach instead of the Eulerian approach of fixing spatial window.

AR: The reviewer is pointing out the critical choice of the reference that is used to perform the rain typing.
As mentioned by the reviewer, in our approach, we adopt an Eulerian reference, i.e. a reference that is fixed with regard to the Earth surface. We think that it is coherent with the target of assessing the temporal stationarity of rainfall statistics at a given location (or over a restricted area) for the purpose of stochastic rainfall modelling.
Because of the choice of this Eulerian reference, the rain type that is 'flying over' the area of interest can change along time due to the travel of distinct storms over the area. This is this succession of rain types that we are trying to identify. This is in line with a large literature about rainfall typing (e.g. Biggerstaff and Listemaa, 2000; Llasat, 2001), with the difference that the number of rain types considered is usually lower, and that the rain types are usually defined from physical properties (e.g. convective vs stratiform rains) while in this study we characterize rain types from their statistical properties.
Therefore our conclusion of sharp transitions between rain types (in an Eulerian reference) does not implies that the nature of a given 'piece of clouds' defined in a Lagrangian reference (i.e. moving with the rain storm) will change along time, but only that distinct 'sub-storms' with distinct features follow each other over the area of interest under the influence of storm advection.
We acknowledge that the question of changing rain types in a Lagrangian reference frame is an interesting and open question, but it is thought to be out of the scope of the present paper. Moreover

we do not know any supporting literature about this point, and we do not aim to show that this phenomenon actually occurs. This would require homogenized radar data over large areas (in any case larger than Switzerland) in order to follow rain storms to track potential changes of their statistics, and for now we do not have access to such data.

To make it unambiguous, we now clearly specify at the end of section 2 (p 5, l 4-5) that we work in an Eulerian reference frame, and that we are interested in the temporal variability of rainfall statistics at a given location.

RC: Are the 10 rainfall characteristics used to parameterize each rainfall imagery independent?

AR: No, we do not assume that the 10 indices used to characterize the rain fields are independent. That is why we use a full covariance model (i.e., accounting for correlations) for the covariance matrices implied in the GMM model, following the guidelines formulated in the review paper by Fraley and Raftery (2002) about GMM clustering. We added a sentence to clarify it (p 7, l 2-3).

RC: I suspect validity of the 10-dimensional Gaussian mixture model because most Copula tend to show awkward behavior as its dimension exceeds 2. Would you show a way to validate the GMM model you developed?

AR: GMM clustering has already been successfully applied to relatively high dimensional problems (more than 10 components). For instance Vrac & Yiou (2010), Rust et al. (2010) or Pernin et al. (2016) applied GMM clustering to problems of dimension 10 to 20.

To make it clear to the reader, we added a reference to the study of Pernin et al. (2016) that uses a high dimensional GMM clustering when we introduce the GMM classifier (p 6, l 23).

RC: I do not believe that the distribution of all 10 variables used for GMM classification has a normal distribution, which is a fundamental model assumption. For example, most rainfall intensity in a time series (in your case, it corresponds to the II-2) has a skewed distribution. You may want to use a type of transfer function to convert the original variables to be normally distributed, and then run the GMM classification algorithm.

AR: The GMM classification does not require normal distributions. Indeed, the idea behind GMM classification is that any multivariate distribution can be approximated by a mixture of multivariate Gaussian distributions if enough components are added. For more insights about GMM classification we refer to the review paper by Fraley and Raftery (2002).

We believe that this misunderstanding originates from our too brief description of the GMM classifier. Therefore, in the new version of our manuscript, the description of the GMM classifier has been improved (from p 6, l 22 to p 7, l 12).

RC: If you cannot validate the previous two points, I suggest you perform the principal component analysis to extract the principle variables, and then apply a simpler clustering algorithms based on the Euclidean distance (e.g. K-means clustering, Hierarchical clustering).

AR: We hope that our responses to the two previous points and the related changes in our manuscript convinced the reviewer that GMM is a rational choice to carry the classification in the specific context of our study.

RC: I believe that the result will be much more stronger if the analysis is performed on the years of the radar rainfall data at multiple locations. I agree with the view of the authors that the absolute validation of the result is not possible because we cannot see the nature thoroughly. However, it cannot be an excuse of not validating your model for a variety of situations.

AR: We agree that for a more exhaustive assessment, our framework should be applied to a wider variety of situations. However, since our method focuses on temporal non-stationarities, we prefer to focus on different seasons rather than on different locations. This is why we added the new section 4.4. to assess our method on a whole year of data (from p 13, l 15 to p 17, l 2). Therefore, multiple meteorological situations are investigated, including winter stratiform rain events, spring showers, and summer convective thunderstorms.

RC: In this view, running a sensitivity analysis will be more helpful. For example, you can change the size of the spatial window, or run the model at different locations with different meteorology, and see how your model behaves.

AR: We agree with the need of a sensitivity analysis to further validate our method, but here again we prefer to focus on the time axis. This is why we added the new section "4.5 Sensitivity of the rain typing approach to the size of the calibration dataset" (from p 17, l 3 to p18, l 12). In this section, rain types are identified based on two calibrations of the GMM classifier: using the data of the summer of 2017 only, and using the data of the whole year 2017. Results show that the two classifications are in relatively good agreement (73.6% of agreement), and that the intra-event rain type transitions are quite clearly identified.

[revised manuscript text omitted]